# DIFFERENTIABLE EXPECTATION-MAXIMIZATION FOR SET REPRESENTATION LEARNING

**Minyoung Kim**
Samsung AI Center Cambridge, UK
`mikim21@gmail.com`

## ABSTRACT

We tackle the set2vec problem, the task of extracting a vector representation from an input set comprised of a variable number of feature vectors. Although recent approaches based on self attention such as (Set)Transformers were very successful due to the capability of capturing complex interaction between set elements, the computational overhead is the well-known downside. The inducing-point attention and the latest optimal transport kernel embedding (OTKE) are promising remedies that attain comparable or better performance with reduced computational cost, by incorporating a fixed number of learnable queries in attention. In this paper we approach the set2vec problem from a completely different perspective. The elements of an input set are considered as i.i.d. samples from a mixture distribution, and we define our set embedding feed-forward network as the maximum-a-posterior (MAP) estimate of the mixture which is approximately attained by a few Expectation-Maximization (EM) steps. The whole MAP-EM steps are differentiable operations with a fixed number of mixture parameters, allowing efficient auto-diff backpropagation for any given downstream task. Furthermore, the proposed mixture set data fitting framework allows unsupervised set representation learning naturally via marginal likelihood maximization aka the empirical Bayes. Interestingly, we also find that OTKE can be seen as a special case of our framework, specifically a single-step EM with extra balanced assignment constraints on the E-step. Compared to OTKE, our approach provides more flexible set embedding as well as prior-induced model regularization. We evaluate our approach on various tasks demonstrating improved performance over the state-of-the-arts.

## 1 INTRODUCTION

Effectively learning succinct and salient feature representation for complex structured data is a central part of recent deep learning. In particular the *set* structured data, arising in many important scientific tasks including bioinformatics and NLP tasks, poses several challenges to deal with varying numbers of elements and the constraints of permutation invariance, in contrast with conventional instance-based representation learning (Zaheer et al., 2017; Edwards & Storkey, 2017; Lee et al., 2019a; Skianis et al., 2020). Perhaps the key to successful set representation learning is to capture the interaction between elements effectively, implemented as self attention (Bahdanau et al., 2015) and (set)transformer models (Vaswani et al., 2017; Lee et al., 2019a).

Despite their superb performance in numerous application problems, one well-known drawback of the self-attention approaches is high computational (time and memory) overhead. Among others, the inducing-point attention (Lee et al., 2019a) and the latest optimal transport kernel embedding (OTKE) (Mialon et al., 2021) are promising remedies that attain comparable or better performance with reduced computational cost. This is enabled by incorporating a fixed number of trainable reference vectors as queries in attention, and especially OTKE, with the attention scheme based on optimal transport, showed state-of-the-art results on several tasks in bioinformatics and NLP.

Although OTKE is motivated from self attention, in this paper we show that OTKE can be derived from a completely different perspective: it is the maximum likelihood estimate of a Gaussian mixture model obtained by a single Expectation-Maximization (EM) step (Dempster et al., 1977) with some constraints on the E-step, which aims to model set elements as i.i.d. data samples. From this

perspective we propose several interesting extension/generalization such as: i) we perform multiple EM steps instead of a single one, ii) unlike treating the references (model parameters) just as an initial iterate of the EM, we also place a prior distribution on the references to incorporate the prior effect to the output embedding in a more principled Bayesian manner. With this generalization, our set embedding function becomes a (differentiable) feed-forward network defined as a multi-step EM for the maximum-a-posterior (MAP) estimate of a Gaussian mixture. Furthermore, the proposed mixture set data fitting framework allows *unsupervised* set representation learning naturally via marginal likelihood maximization aka the empirical Bayes.

On several set/sequence representation learning problems including biological sequence classification problems and NLP tasks, our approach shows considerable improvement over the state-of-the-arts including OTKE. Our main contributions are summarized below:

1. We derive OTKE by a novel EM-based mixture fitting perspective, which allows us to generalize it to a more flexible and principled set embedding function that can offer more modeling options, prior-induced model regularization, and unsupervised learning via empirical Bayes.

2. The whole MAP-EM steps are differentiable operations with a fixed number of mixture parameters, enabling efficient auto-diff back-propagation for any given downstream task.

3. Our approach shows improved results over state-of-the-arts on various set embedding tasks in bioinformatics and NLP that require effective interaction modeling among set elements.

## 2 BACKGROUND ON OTKE

OTKE (Mialon et al., 2021) is a computationally efficient trainable model for representing a variable-length set, by aggregating a variable number of features/elements in a set into a fixed-size vector. The input to the model is a set $S = \{x_1, \ldots, x_n\}$ ($n$ varies from instance to instance), where each $x_i \in \mathbb{R}^d$ is an input feature vector which may be an output of some feature extractor network, thus involving some trainable parameters in it. For instance, in the OTKE paper, they often consider a parametric Nyström approximation of the RKHS $\phi(x)$ (Williams & Seeger, 2001). We abuse the notation throughout the paper, denoting by $x$ both a raw input element and an extracted feature $\phi(x)$, where the distinction will be clear from context.

The OTKE model's set embedding function, $emb : \mathbb{S} \to \mathbb{R}^D$ where $\mathbb{S} \ni S$, is defined as follows:

$$emb(S) = \sqrt{p} \cdot \text{cat}\left( \sum_{i=1}^{n} Q_{i1} x_i, \cdots, \sum_{i=1}^{n} Q_{ij} x_i, \cdots, \sum_{i=1}^{n} Q_{ip} x_i \right) \qquad (1)$$

where cat() indicates concatenation of vectors in column, and hence the output is of dimension $D = d \cdot p$. The trainable parameters of the embedding function are the $p$ *reference vectors* $Z = \{z_1, \ldots, z_p\}$ with $z_j \in \mathbb{R}^d$, where they serve as *queries* to determine the weights $Q$ in the attention form (1). Instead of following the popular dot-product strategy (Vaswani et al., 2017), the weight matrix $Q$ is the (unique) solution of the optimal transport problem (Villani, 2008; Cuturi, 2013) between $S$ and $Z$ with the cost matrix $C$ defined as the negative kernel $C_{ij} = -k(x_i, z_j)$, that is,

$$\min_Q \; \sum_{ij} C_{ij} Q_{ij} - \epsilon H(Q) \; \text{ s.t. } \; \sum_{i=1}^{n} Q_{ij} = \frac{1}{p}, \; \forall j \; \text{ and } \; \sum_{j=1}^{p} Q_{ij} = 1/n, \; \forall i \qquad (2)$$

where the objective is augmented with the entropic term (impact $\epsilon$), $H(Q) = -\sum_{ij} Q_{ij} \log Q_{ij}$, allowing the efficient Sinkhorn-Knopp (SK) algorithm to be applied (Cuturi, 2013).

This OT-based weighting is shown to be more effective than the dot-product in the biological application domain (Mialon et al., 2021). The key difference from dot-product is the balanced assignment constraints $\sum_i Q_{ij} = 1/p$ for all $j = 1, \ldots, p$, and the main motivation for this is to have all references $z_j$ contribute equally to describing the set $S$. This may be a reasonable strategy in certain scenarios to prevent a small number of references from affecting the attention dominantly.

Note that (2) can be solved by a few fixed point matrix scaling iterations (SK algorithm): The SK algorithm finds the optimal solution as $Q = \text{Diag}(u)A\text{Diag}(v)$, where $A_{ij} = \exp(-C_{ij}/\epsilon)$ and the vectors $u \in \mathbb{R}^n_+$ and $v \in \mathbb{R}^p_+$ are the fixed points of $u_i = \frac{1}{n}/(Av)_i$, $v_j = \frac{1}{p}/(A^\top u)_j$ for $i = 1, \ldots, n$,

$j = 1, \ldots, p$. The fixed point iteration usually converges quickly after a few iterations. Since the whole SK iterations constitute the feed-forward pass for the final output (1), hence OT is differentiable. Note that as the transport cost in OT is defined as the negative kernel, it becomes simply the dot-product in the RKHS, i.e., $k(x, z) = x^\top z$ (with the RKHS embedding notation, $\phi(x)^\top z$), where $z$ lives in the same RKHS. OTKE can be extended to a multi-head network, for which we run the above OTKE procedure multiple times independently, each with different parameters $Z$, then concatenate the outputs.

OTKE has several notable advantages over the self attention. First, it is computationally efficient, scalable to very large dataset. Unlike the quadratic cost of the self attention, OTKE uses a fixed number ($p$) of references serving as queries. Secondly, it allows unsupervised learning where the model parameters $Z$ can be learned without target (class) labels via the Wasserstein (or k-means alternatively) clustering of set elements as per the OT problem. This unsupervised learning can also serve as pre-training for the potentially unknown target downstream tasks. We also note that regarding the first benefit, the formula (1) is also very similar to the inducing-point attention of SetTransformer (Lee et al., 2019a) (known as the PMA layer) except for the OT-based weighting scheme. In Appendix B.2 we provide a proof showing that PMA is indeed a special case of OTKE.

## 3 EXPECTATION MAXIMIZATION VIEW OF OTKE

While OTKE is motivated and derived from the attention mechanism, in this section we derive the OTKE embedding formulas (1-2) from a completely different perspective, namely the EM-based maximum likelihood learning of the Gaussian mixture for set data. Specifically we consider a mixture of $p$ Gaussians with equal mixing proportions and fixed shared spherical covariances, that is,

$$p(x|\theta) = \sum_{j=1}^{p} \underbrace{(1/p)}_{p(c=j)} \underbrace{\mathcal{N}(x; z_j, \epsilon I)}_{p(x|c=j)} \tag{3}$$

where $c$ denotes the component latent variable, and the learnable parameters are only the means $\theta = \{z_1, \ldots, z_p\}$. By regarding the set elements $x_i$'s as i.i.d. data samples, the log-likelihood of the set $S = \{x_1, \ldots, x_n\}$ under the model is $\log p(S|\theta) = \sum_i \log p(x_i|\theta)$.

Now we perform *one* E-step in the EM toward maximizing the log-likelihood, which is derived by the following Jensen lower bound:

$$\log p(S|\theta) = \sum_{i=1}^{n} \log \sum_{j=1}^{p} p(x_i, c_i = j) \geq \sum_{i=1}^{n} \sum_{j=1}^{p} q(j|i) \log \frac{p(x_i, c_i = j)}{q(j|i)} \tag{4}$$

where $q(j|i)$ is the variational distribution and the bound is tight if $q(j|i) = p(c_i = j|x_i)$. Although the latter can be derived in a closed form, we aim to formulate an optimization problem for $q(j|i)$. We use a joint variational form instead, i.e., $q_{ij} := q(j|i) \cdot q(i)$ with $q(i) := 1/n$. Then maximizing the lower bound in (4) with respect to the ($n \times p$) matrix $q = (q_{ij})$ is equivalent to solving:

$$\min_q \sum_{ij} C'_{ij} q_{ij} - H(q), \quad \text{where} \quad C'_{ij} = -\log p(x_i, c_i = j) = \frac{||x_i - z_j||^2}{2\epsilon} + \text{const.} \tag{5}$$

By having the quadratic kernel $k(x, z) = -\frac{1}{2}||x - z||^2$ which is (conditionally) positive definite (Schölkopf & Smola, 2002), we see that (5) reduces to (2) except for the first constraints $\sum_i q_{ij} = 1/p$ (the other constraints $\sum_j q_{ij} = 1/n$ are automatically met by construction). Note that the quadratic kernel approximates the dot-product (in the RKHS) if the norms of the points are roughly constant.

Hence the conventional E-step of the EM algorithm coincides with the OT-based attention weight computation (2) without the balanced assignment constraints $\sum_i q_{ij} = 1/p$. Conversely, if we consider to maximize the lower bound (4) with the extra balanced assignment constraints on the variational distribution $q = (q_{ij})$, the corresponding E-step exactly reduces to the OT solution (2).

Once the E-step is done (i.e., $q$ is found), we perform the M-step, that is, maximizing the lower bound in (4) with respect to $\theta$ while $q$ is fixed. This admits a closed-form update equation,

$$z_j^{new} = \frac{\sum_{i=1}^{n} q_{ij} x_i}{\sum_{i=1}^{n} q_{ij}} \quad (j = 1, \ldots, p). \tag{6}$$

Note that if we imposed the balanced assignment constraints, the denominator of (6) is constant $(1/p)$, and by concatenating $z_j^{new}$'s, it exactly reduces to the OTKE formula (1) (up to a constant factor). But, even without the constraints, the M-step in (6) suggests to normalize (balance) the scales of the transformed vectors $\sum_i q_{ij} x_i$ across $j = 1, \ldots, p$, according to the reference marginals $\sum_i q_{ij}$ (how many points are assigned to $z_j$).

This establishes a very close connection between OTKE and the EM-updated mean parameters of the Gaussian mixture. Note that our derivation showed that OTKE is recovered by just *one* EM step, and there is possibility of extending it by taking multiple EM steps. Furthermore, this opens up other interesting generalization of OTKE through the (EM) mixture set data fitting perspective. We summarize below reasonable and interesting generalization/extension of OTKE in four folds:

1. Instead of fixing mixing proportions and (shared) covariances in the Gaussian mixture as in (3), we can also incorporate them as parameters of the model in conjunction with the means.

2. Instead of just one EM step, we can take multiple ($k$) EM steps with $k$ as a hyperparameter.

3. It becomes an option whether the E-step is solved with the balanced assignment constraints (solving the OT problem) or not (regular E-step). This enables more flexible modeling.

4. OTKE can be seen as an (single-step) EM which starts from the references $z_j$'s as an initial iterate. Not just using them as an initial iterate, we can also place a prior distribution on the Gaussian mixture parameters, that is, $p(\theta)$ which serves as learnable model parameters. This allows us to define the embedding as the posterior distribution $emb(S) = p(\theta|S)$.

These extensions of OTKE are incorporated into a single framework, which becomes a novel set embedding function viewed as a differentiable MAP Expectation-Maximization for a Gaussian mixture.

## 4 DIFFERENTIABLE EM (DIEM)

We consider a Gaussian mixture model with non-fixed mixing proportions and covariances:

$$p(x|\theta) = \sum_{j=1}^{p} \pi_j \mathcal{N}(x; z_j, V_j), \quad \theta = \{\pi_j, z_j, V_j\}_{j=1}^{p} \qquad (7)$$

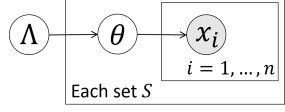

Figure 1: DIEM's graphical model diagram.

where the covariances $V_j$ are typically chosen as diagonal $(d \times d)$ matrices[1] to reduce the computational overhead. Given the mixture $\theta$, the set elements are considered i.i.d., $p(S|\theta) = \prod_{i=1}^{n} p(x_i|\theta)$. Now we impose a prior distribution on $\theta$ where the Dirichlet-Normal-Inverse-Wishart (*Dir-NIW* for short) would be a fairly natural choice:

$$p(\theta|\Lambda) = Dir(\pi; \eta) \cdot \prod_{j=1}^{p} \mathcal{NIW}(z_j, V_j; \mu_j, \lambda, \Sigma_j, \nu), \quad \Lambda = \{\eta, \lambda, \nu, \{\mu_j, \Sigma_j\}_{j=1}^{p}\}. \qquad (8)$$

This defines a graphical model (whose diagram shown in Fig. 1) where $\Lambda$ generates mixture parameters $\theta$'s, one for each set $S$, and each $\theta$ generates $x_1, \ldots, x_n \in S$. We emphasize that the reason we consider the prior is that we can express the process of how the model parameters impact on the final output in a more principled manner, beyond just having them as an initial iterate. For simplicity, the prior parameters $\eta, \lambda, \nu$ are shared across the components. That is, $Dir(\pi; \eta) \propto \prod_j \pi_j^{\eta-1}$ where $\eta$ is scalar, and $\mathcal{NIW}(z_j, V_j) = \mathcal{N}(z_j; \mu_j, V_j/\lambda) \cdot \mathcal{IW}(V_j; \Sigma_j, \nu)$ for all $j = 1, \ldots, p$, where $\lambda, \nu$ are scalar, $\mu_j \in \mathbb{R}^d$, and $\Sigma_j$ are positive diagonal matrices.

Then we define our set embedding function $emb(S) \to \mathbb{R}^D$ as the MAP estimate of the Gaussian mixture $\theta$, denoted by $\theta^{MAP}(S) := \arg\max_\theta p(\theta|S, \Lambda)$, which is approximated by the $k$-step EM update. And the prior parameters $\Lambda$ will constitute the model parameters of the embedding function $emb(S)$ to be trained. The EM update equations for the MAP can be derived in closed forms. More specifically, from the following lower-bound of the log-posterior ("$=_c$" means equal up to constant)

$$\log p(\theta|S, \Lambda) =_c \log p(\theta|\Lambda) + \log p(S|\theta) \geq \log p(\theta|\Lambda) + \sum_{ij} q(j|i) \log \frac{p(x_i, j)}{q(j|i)}, \qquad (9)$$

---

[1]In some of our experiments in Sec. 6, we used fixed spherical covariances instead which performed better.

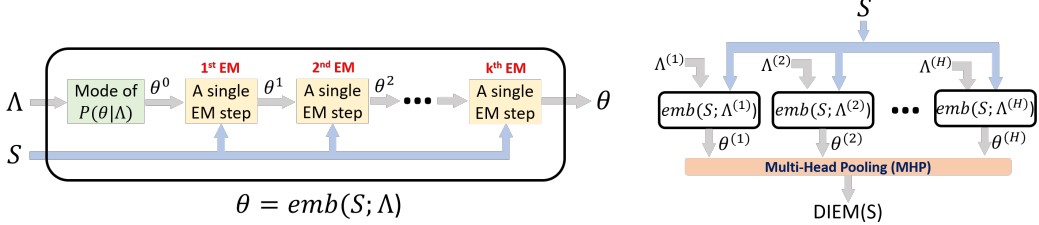

Figure 2: Network diagrams of our Differentiable EM (DIEM) model. (Left) $\theta = emb(S; \Lambda)$ as a feed-forward network that performs the $k$-step MAP EM starting from the mode of the prior $\theta^0$. (Right) Multi-head representation by MHP where we propose three strategies. See text for details.

the E-step maximizes the lower bound with respect to $q$ (with $\theta$ fixed), which yields

$$\textbf{E-step}: \ q(j|i) = \frac{\pi_j \mathcal{N}(x_i; z_j, V_j)}{\sum_j \pi_j \mathcal{N}(x_i; z_j, V_j)} \ \ \text{or} \ \ q(j|i) = n \cdot \text{SK}(C = \{-\log p(x_i, j)\}, \epsilon = 1). \quad (10)$$

The former formula in (10) is the regular E-step without the balanced assignment constraints, and the latter is the solution with the constraints where $\text{SK}(C, \epsilon)$ indicates the solution of the OT problem via the SK algorithm (c.f., (2) and (5)). Note that we multiply the latter by $n$ to have conditional distributions $q(j|i)$.

The M-step is the log-prior regularized weighted log-likelihood maximization with respect to $\theta$ (with $q$ fixed), which admits a closed-form solution (Appendix B.1 for derivations)

$$\textbf{M-step}: \ \pi'_j = \frac{\sum_i q(j|i) + \tau_\eta}{n + p \cdot \tau_\eta}, \quad z'_j = \frac{\sum_i q(j|i)x_i + \mu_j \tau_\lambda}{\sum_i q(j|i) + \tau_\lambda},$$

$$V'_j = \frac{\sum_i q(j|i)(x_i^2 - (z'_j)^2) + \Sigma_j \tau_1 + (\mu_j^2 - (z'_j)^2)\tau_\lambda}{\sum_i q(j|i) + \tau_\nu} \quad (11)$$

where all operations including squaring are element-wise. The $\tau$ values originate from the prior, defined as: $\tau_\eta = \eta - 1$, $\tau_\lambda = \lambda$, $\tau_1 = 1$, and $\tau_\nu = \nu + d + 2$. Note that $\tau$'s represent the strength of the prior impact: turning them off to $0$ completely ignores the prior and leads to the maximum likelihood M-step update. Instead of dealing with them differently, we simplify it by collapsing them into a single hyperparameter $\tau$ $(= \tau_\eta = \tau_\lambda = \tau_1 = \tau_\nu)$. Then the **simplified M-step** becomes:

$$\pi'_j = \frac{\sum_i q(j|i) + \tau}{n + p \cdot \tau}, \ z'_j = \frac{\sum_i q(j|i)x_i + \mu_j \tau}{\sum_i q(j|i) + \tau}, \ V'_j = \frac{\sum_i q(j|i)x_i^2 + (\Sigma_j + \mu_j^2)\tau}{\sum_i q(j|i) + \tau} - (z'_j)^2. \quad (12)$$

Now we can control $\tau$ to impose the prior impact in the set embedding: smaller (larger) $\tau$ for lower (higher, resp.) prior impact. We treat $\tau$ as the hyperparameter (e.g., chosen by validation) while $\Lambda = \{\mu, \Sigma\}$ constitutes the learnable prior parameters.

As an initial iterate $\theta^0$ of the EM, we use the mode of prior $p(\theta|\Lambda)$, which equals

$$\textbf{Initial } \theta^0: \ \pi_j^0 = \frac{1}{p}, \quad z_j^0 = \mu_j, \quad V_j^0 = \frac{\Sigma_j}{\tau + 1}. \quad (13)$$

Note that $\theta^0$ is a function of $\Lambda$. We deal with multiple ($k$) EM steps, $\theta^0 \to \theta^1 \to \cdots \to \theta^k$, each following E-step (10) and M-step (12), and the final $\theta^k$ is a function of $\Lambda$. Overall, from $\Lambda$ to the final iterate $\theta^k$ constitutes a feed-forward pass, and it is composed of all differentiable operations. The final output $emb(S; \Lambda) = \theta^k \approx \theta^{MAP}(S; \Lambda)$ is then transformed to a vector representation by concatenating the parameters in $\theta^k$ in a fixed order (e.g., $\text{cat}(\pi_1^k, z_1^k, V_1^k, \pi_2^k, z_2^k, V_2^k, \ldots, \pi_p^k, z_p^k, V_p^k)$). The feed-forward network of $emb(S; \Lambda)$ is visualized in Fig. 2 (Left).

**Multi-head representation.** We can also extent the model to have multi-head representation, which is simply done by replicating the process $H$ times with different prior parameters $\Lambda^1, \ldots, \Lambda^H$. See Fig. 2 (Right). We have $H$ outputs from the MAP-EM set embedding networks, denoted by $\theta^{(h)}$ for $h = 1, \ldots, H$. Then the final output is a pooling from these $\theta^{(h)}$'s, and we propose three different options for this multi-head pooling (MHP): i) **PC** (parameter-concatenation) simply concatenates $\theta^{(h)}$'s in a fixed order, ii) **SB** (select-best) computes the marginal data likelihood $\log p(S|\Lambda^h)$ for each $h$ and select/return the one with the largest score only, and iii) **SB2** outputs the additional one-hot encoding of the selected mixture's ID in addition to SB's output. Note that **PC** makes sense in that we have the supports of the input set covered by all $H$ mixtures together, while **SB** strategy is motivated

from the conventional practice in EM learning where we run EM algorithm multiple times with differential initial iterates and take the one with the highest likelihood score. **SB2** additionally links the output to the specific mixture parameters selected.

**Unsupervised learning.** DIEM can be typically used in supervised learning, where the output of $DIEM(S)$ is fed into subsequent layers (including the classification head) in a prediction network, and updated in an end-to-end learning manner. Beyond this, DIEM can be trained with only sets $\{S_1, \ldots, S_m\}$ without target labels. This is naturally done by marginal data likelihood maximization for mixture learning, where we maximize $\sum_{i=1}^{m} \log p(S_i)$ with respect to the prior parameters $\Lambda$. We can perform EM for optimizing the objective (in this case, the off-line EM until convergence with the entire training sets, without the auto-diff backprop mode), or alternatively the k-means clustering. The latter leads to the unsupervised learning algorithm similar to that of OTKE (Mialon et al., 2021).

## 5 RELATED WORK

**Set representation learning.** Prior to the attention-based approaches, the neural statistician (Edwards & Storkey, 2017) aims to build a model that learns the statistics of an input set, while DeepSet (Zaheer et al., 2017) has the element-wise nonlinear transformation followed by a simple mean or max pooling to aggregate features. Although the universality of this network architecture was partially proved, the limitation was found (Wagstaff et al., 2019), and network structures more suitable for exchangeability and permutation invariance were suggested (Garnelo et al., 2018; Bloem-Reddy & Teh, 2019).

**Attention-based methods.** Self attention was shown to be very successful for modeling set-structured data (Vinyals et al., 2016; Yang et al., 2018; Ilse et al., 2018; Kim et al., 2019). While there were several sophisticated attempts to reduce the quadratic computational cost of self attention (Wang et al., 2020; Kitaev et al., 2020), the inducing-point approaches that introduce learnable parameters as queries are especially suitable for set inputs (Lee et al., 2019a; Pritzel et al., 2017; Skianis et al., 2020; Mialon et al., 2021). The attention mechanism was analyzed rigorously in (Tsai et al., 2019) providing unifying kernel interpretation. Applying multiple EM steps in our DIEM is architecturally similar to the recent Perceiver model (Jaegle et al., 2021) with multiple latent-attention layers.

**Mixture-based representation.** There was an attempt to define the similarity between two images (image as a set of fixed feature vectors, e.g., Fisher vectors (Perronnin et al., 2010)) using mixture models about a decade ago (Liu & Perronnin, 2008). Similar to ours, they represent each image as a MAP estimated mixture density, and the similarity between two images is measured as a divergence between the corresponding mixture densities. However, instead of treating the pipeline of whole EM steps as a differentiable object and learning the prior parameters as in our DIEM, they merely apply a single MAP-EM step to obtain a fixed density for each image, where the initial mixture for EM is also fixed from global offline learning with entire training images.

## 6 EVALUATION

We evaluate our DIEM model empirically on two different types of tasks: i) counting and clustering problems (Sec. 6.1, 6.2) to verify the model's capability of learning general set representations by modeling interaction between set elements, ii) large-scale biological sequence classification and NLP tasks (Sec. 6.3, 6.4, 6.5) to test the performance of the proposed model on real-world problems in both supervised and unsupervised settings.

The hyperparameters in our DIEM include: $p$ (the mixture order), $H$ (the number of heads), $k$ (the number of EM steps), $\tau$ (prior impact), and the multi-head pooling strategy (either of PC, SB, or SB2). We report the results of the best combinations that are selected by cross validation. The empirical study on the impact of these hyperparameters is summarized in Appendix C.4. The other option is the mixture modeling: we learn both means and covariances for counting, clustering, and SCOP 1.75 datasets, while covariances are fixed for SST-2 and DeepSEA.

### 6.1 OMNIGLOT UNIQUE CHARACTER COUNTING

This task, originally devised in (Lee et al., 2019a), is useful to verify capability of modeling interaction between set elements. From the OMNIGLOT dataset (Lake et al., 2015) which contains 1,623

Table 1: Unique character counting on OMNIGLOT. The averaged 0/1 accuracy results.

| Method | Small set | Large set |
|---|---|---|
| DeepSet (Zaheer et al., 2017) | $0.4617 \pm 0.0076$ | $0.1927 \pm 0.0091$ |
| Dot-prod Attn (Yang et al., 2018; Ilse et al., 2018) | $0.4471 \pm 0.0076$ | N/A |
| SetTransformer (SAB + PMA) (Lee et al., 2019a) | $0.6037 \pm 0.0075$ | $0.3191 \pm 0.0050$ |
| OTKE (Mialon et al., 2021) | $0.5754 \pm 0.0130$ | $0.3352 \pm 0.0098$ |
| DIEM (Ours) | $\mathbf{0.7153 \pm 0.0067}$ | $\mathbf{0.4440 \pm 0.0069}$ |

different handwritten characters with 20 examples each, we build a set by randomly selecting $n$ images, and the task is to predict the number of unique characters in the set. Following the data construction protocol from (Lee et al., 2019a), we split the 1,623 characters into training/validation/test splits so that the test dataset contains only unseen characters. Specifically, we randomly choose the number of images $n$ uniformly from $\{N_{min}, \ldots, N_{max}\}$, then choose the number of unique characters $c$ randomly from $\{c_{min}, \ldots, n\}$. Then we randomly sample $c$ different classes (characters), and sample $n$ images from those classes while ensuring that all classes have at least one image. We form two experimental settings: i) Small set ($N_{min} = 6, N_{max} = 10, c_{min} = 1$) originally used in (Lee et al., 2019a) and ii) (a more challenging) Large set ($N_{min} = 11, N_{max} = 30, c_{min} = 5$).

For the models, we adopt a similar feature extractor architecture as (Lee et al., 2019a): first apply four Conv-BN-ReLU layers to each (element) image to have feature representation $\phi(x)$, then perform the set embedding $emb(S)$ whose output is fed into a fully connected layer to return the Poisson parameter $\lambda$. The final loss function is the negative Poisson log-likelihood $-\log p(c|\lambda)$. The batch size is 32 (sets) for both datasets.

**Results.** We report the 0/1 prediction accuracy averaged over 10 random runs in Table 1. OTKE, after validation, uses ($p = 100, H = 1$) for the small set and ($p = 50, H = 2$) for the large set. Our DIEM takes ($p = 20, H = 3, k = 2, \tau = 10^{-3}$, PC) for the small set and ($p = 50, H = 2, k = 2, \tau = 10^{-3}$, PC) for the large set. For both datasets, our DIEM outperforms the competing approaches by large margin. Unlike OTKE, we have prior regularization and multiple EM steps which appear to be important to improve the performance.

## 6.2 AMORTIZED CLUSTERING

The task of amortized clustering is to learn a function $f(S) \to \Theta$ where $S = \{x_1, \ldots, x_n\}$ is a (data) set of $n$ points, and $\Theta = \{\alpha, m, C\}$ is the parameters of the mixture of Gaussians, $p(x; \Theta) = \sum_j \alpha_j \mathcal{N}(x; m_j, C_j)$, best fit to the data. We follow the setup similar to (Lee et al., 2019a) where instead of providing the supervised data of pairs[2] $\{(S, \Theta)\}$, we are only given the set data $S$, and the loss function is the negative log-likelihood $-\log p(S|\Theta)$. Following (Lee et al., 2019a), we form two datasets: i) Synthetic dataset where $n$ 2D data points are sampled from a randomly generated mixture of four Gaussians, and ii) CIFAR-100 images where we first sample four classes out of 100, and sample $n$ images from the classes to form a set $S$. In both cases $n \in \{100, \ldots, 500\}$. For CIFAR-100, each image is represented as a 512-dim vector from a VGG network pre-trained on the training set (Simonyan & Zisserman, 2014).

For our DIEM (and OTKE), we first apply a simple linear layer transformation to have features $\phi(x)$. Then the output of the set embedding $emb(S)$ is fed into a fully connected layer to have the Gaussian mixture parameters $\Theta$. For the competing DeepSet and SetTransformer models, we faithfully follow the network structures in (Lee et al., 2019a). The details are summarized in Appendix C.2.

**Results.** The results are reported in Table 2 where the test log-likelihood scores and the adjusted rand index (ARI) are shown for the synthetic and CIFAR-100 datasets, respectively. As references, we also report the oracle performances: (Synthetic) the true mixture used to generate the data, and (CIFAR-100) the full offline EM-trained mixture model. Although SetTransformer greatly outperforms DeepSet, our DIEM exhibits further improvement over SetTransformer by large margin. Relatively poor performance of OTKE may be due to only one-step EM update, compared to our DIEM that used $k = 3$ EM steps. For instance, DIEM also showed degraded performance when $k = 1$ is used (Fig. 10 in Appendix C.4). Another reason of success of DIEM is the prior-induced regularization ($\tau = 0.01$) whereas OTKE has no such regularization (Fig. 11 in Appendix C.4).

---

[2]This is the typical setup for the research topic called *deep amortized clustering* (Lee et al., 2019b; Pakman et al., 2020; Genevay et al., 2019), which is different from our general set representation learning.

Table 2: Amortized clustering on 2D synthetic data and CIFAR-100. For CIFAR-100, since it was difficult to obtain exactly the same pre-trained VGG network as in (Lee et al., 2019a), we ran all models with our own pre-trained VGG network (about 71% test accuracy compared to 68.54% from (Lee et al., 2019a)).

| Method | Synthetic | CIFAR-100 |
|---|---|---|
| | Log-likelihood $\uparrow$ | Adjusted rand index $\uparrow$ |
| Oracle | -1.4726 | 0.9842 |
| DeepSet (mean pooling) (Zaheer et al., 2017) | $-1.7606 \pm 0.0213$ | $0.5736 \pm 0.0117$ |
| DeepSet (max pooling) (Zaheer et al., 2017) | $-1.7692 \pm 0.0130$ | $0.5463 \pm 0.0154$ |
| Dot-prod Attn (Yang et al., 2018; Ilse et al., 2018) | $-1.8549 \pm 0.0128$ | N/A |
| SetTransformer (SAB + PMA) (Lee et al., 2019a) | $-1.5145 \pm 0.0046$ | $0.9246 \pm 0.0113$ |
| SetTransformer (ISAB16 + PMA) (Lee et al., 2019a) | $-1.5009 \pm 0.0068$ | $0.9381 \pm 0.0122$ |
| OTKE ($p = 4, H = 5$) (Mialon et al., 2021) | $-1.7803 \pm 0.0028$ | $0.8207 \pm 0.0074$ |
| DIEM ($p = 4, H = 5, k = 3, \tau = 0.01$, SB2) (Ours) | $\mathbf{-1.4873 \pm 0.0018}$ | $\mathbf{0.9770 \pm 0.0019}$ |

Table 3: SCOP 1.75 classification accuracies (top 1/5/10) for unsupervised and supervised learning.

| Method | Unsupervised | Supervised |
|---|---|---|
| DeepSF (Hou et al., 2019) | N/A | 73.0 / 90.3 / 94.5 |
| CKN (Chen et al., 2019a) | $81.8 \pm 0.8$ / $92.8 \pm 0.2$ / $95.0 \pm 0.2$ | $84.1 \pm 0.1$ / $94.3 \pm 0.2$ / $96.4 \pm 0.1$ |
| RKN (Chen et al., 2019b) | N/A | $85.3 \pm 0.3$ / $95.0 \pm 0.2$ / $96.5 \pm 0.1$ |
| SetTransformer (Lee et al., 2019a) | N/A | $79.2 \pm 4.6$ / $91.5 \pm 1.4$ / $94.3 \pm 0.6$ |
| Rep-the-Set (Skianis et al., 2020) | N/A | $84.5 \pm 0.6$ / $94.0 \pm 0.4$ / $95.7 \pm 0.4$ |
| OTKE (Mialon et al., 2021) | $85.8 \pm 0.2$ / $95.3 \pm 0.1$ / $96.8 \pm 0.1$ | $88.7 \pm 0.3$ / $95.9 \pm 0.2$ / $97.3 \pm 0.1$ |
| DIEM (Ours) | $\mathbf{86.4 \pm 0.1}$ / $\mathbf{95.6 \pm 0.1}$ / $\mathbf{97.1 \pm 0.1}$ | $\mathbf{90.5 \pm 0.2}$ / $\mathbf{96.6 \pm 0.2}$ / $\mathbf{97.6 \pm 0.2}$ |

## 6.3 PROTEIN FOLD CLASSIFICATION TASK ON SCOP 1.75

Protein fold classification is the well-known task in bioinformatics where the goal is to predict the fold class for a given protein sequence. We use the preprocessed data of the SCOP version 1.75 and 2.06 from (Hou et al., 2019; Mialon et al., 2021), which consists of 19,245 sequences (14,699/2,013 training/validation from SCOP 1.75 and 2,533 test from SCOP 2.06). Each input protein is a sequence of amino acids whose length ranges from tens to thousands, and each amino acid is represented as a 45-dimensional vector based on the PSSM, secondary structure, and more. It is a multi-classification task to predict one of the 1,195 different folds to which a protein sequence belongs.

For the models, we adopt the overall network architecture from (Chen et al., 2019a; Mialon et al., 2021): each input sequence is treated as a set $S$, each set element $\phi(x)$ is a learnable Gaussian kernel mapping on 10-mers via Nyström RKHS approximation with $L$ anchor points (we use $L = 1024$ for unsupervised learning, and $L = 512/128$ for supervised learning), and the output of the set embedding $emb(S)$ is fed into the final softmax linear classification layer. For the competing methods, we replace the set embedding $emb(S)$ by: the global mean pooling layer for CKN (Chen et al., 2019a), the OTKE layer (Mialon et al., 2021), and our DIEM layer. In the unsupervised learning, we sequentially train-then-fix the first two layers without class labels: train $\phi(x)$ via k-means, fix it, do unsupervised training of $emb(S)$, and fix it, after which the last classification layer is trained with class labels. We use the k-means clustering for unsupervised learning. In the supervised learning case, the whole network is trained end-to-end.

**Results.** Classification accuracy (top 1/5/10) for unsupervised and supervised learning are shown in Table 3. In our DIEM, we fix the covariances $V$ as identity and perform $k = 1$ OT E-step for the unsupervised learning, while we learn priors for $V$ and take $k = 2$ regular E-steps for the supervised learning. The details of the hyperparameters are summarized in Table 12 of Appendix C.3. The result shows that our approach outperforms all state-of-the-arts for both learning settings. Compared to OTKE, the increased accuracy of our DIEM can be mainly attributed to the prior-induced model regularization ($\tau = 10^{-3}$ for unsupervised learning) and multiple EM steps ($k = 2$ for supervised learning). See also Fig. 12 in the Appendix for the hyperparameter impact analysis.

## 6.4 NLP TASK: SENTIMENT CLASSIFICATION ON SST-2

The dataset (Socher et al., 2013) consists of 70,042 movie reviews with positive/negative binary sentiment. We follow the experimental protocol from (Mialon et al., 2021) where the original 67,349

Table 4: SST-2 sentiment classification accuracies for unsupervised and supervised learning.

| Method | Unsupervised | Supervised |
|---|---|---|
| BERT [CLS] embedding (Devlin et al., 2019) | $84.6 \pm 0.3$ | $90.3 \pm 0.1$ |
| BERT mean pooling (Devlin et al., 2019) | $85.3 \pm 0.4$ | $\mathbf{90.8 \pm 0.1}$ |
| SetTransformer (Lee et al., 2019a) | N/A | $87.9 \pm 0.8$ |
| Approximate Rep-the-Set (Skianis et al., 2020) | N/A | $86.8 \pm 0.9$ |
| Rep-the-Set (Skianis et al., 2020) | N/A | $87.1 \pm 0.5$ |
| OTKE (Mialon et al., 2021) | $86.8 \pm 0.3$ | $88.1 \pm 0.8$ |
| DIEM (Ours) | $\mathbf{87.6 \pm 0.2}$ | $88.7 \pm 0.4$ |

training data are split into $80\%/20\%$ training/validation sets, and the 872 validation reviews form a test set. This is because test evaluation on the original test set requires online submission in the GLUE (Wang et al., 2019) leaderboard. Similar to (Mialon et al., 2021), we adopt the 768-dimensional word vectors from the pre-trained BERT model (Devlin et al., 2019; Wolf et al., 2019), and use the Gaussian RKHS mapping $\phi(x)$ approximated by the Nyström method with 2048 (unsupervised) or 64 (supervised) filters. Overall network architectures and learning options are also similar to (Mialon et al., 2021).

**Results.** Table 4 summarizes the results. For the unsupervised learning, the proposed DIEM attains the best accuracy, even outperforming the pre-trained BERT model with the popular [CLS] embedding or mean pooling. Unlike OTKE that takes a large number of parameters ($p = 300, H = 1$), our DIEM has far fewer parameters ($p = 3, H = 2, k = 1$ with PC multi-head pooling). When we trained OTKE with the same complexity architecture, it only attained accuracy $86.6 \pm 0.5$. The improved accuracy of DIEM over OTKE is mainly due to the prior-induced regularization (we use $\tau = 10^{-3}$) and the regular E-step (with the OT E-step, the performance is on par, $86.3 \pm 0.8$). For the supervised learning, both DIEM and OTKE has $p = 30, H = 4$, and the performance difference mainly comes from the regular E-step taken by DIEM ($\tau = 10^{-6}$, SB). Although DIEM performs comparably well, it falls short of the fine-tuned BERT model. As alluded in (Mialon et al., 2021), the set embedding approaches may not be very attractive in this case due to short sentences (small cardinality sets).

### 6.5 CHROMATIN PROFILE DETECTION ON DEEPSEA

Finally, we test our DIEM on the large-scale DeepSEA dataset (Zhou & Troyanskaya, 2015) with about five million genomic sequences. It is a multi-label classification task to predict 919 chromatin profiles. As in (Mialon et al., 2021), the overall network architecture follows the 1D Conv layer from DeepSEA, while the pooling layer after the Conv is replaced by the DIEM layer. The RKHS feature mapping is

Table 5: Chromatin profile detection results on DeepSEA.

| Method | auROC | auPRC |
|---|---|---|
| DeepSEA | 0.933 | 0.342 |
| OTKE | 0.936 | 0.360 |
| DIEM (Ours) | 0.936 | 0.360 |

not used (identity mapping), and the Gaussian positional encoding[3] was adopted. The supervised learning results are shown in Table 5, where we have marginal improvement over DeepSEA, and the same performance as OTKE using the ($p = 64, H = 1, k = 2, \tau = 1$) hyperparameter choice.

## 7 CONCLUSION

In this paper we proposed a novel differentiable EM model for set representation learning. Our model is built from the perspective of fitting a Gaussian mixture model to the set data that are viewed as i.i.d. samples, which offers more flexibility and prior-induced model regularization in a principled Bayesian manner. The proposed model is also shown to generalize the recent set embedding models based on optimal transport and attention, leading to a computationally efficient model with superb performance on tasks in bioinformatics and NLP. The mixture fitting perspective can potentially solve other important problems in set representation learning. For instance, deciding the number of components/references can be translated into the mixture order selection problem, and tackled by well-known information criteria methods. This will be left as interesting future research.

---

[3]As in (Mialon et al., 2021), the cost matrix in OT is multiplied by the RBF-like position similarity kernel $P_{ij} = \exp(-(i/n - j/p)/\sigma^2)$. Alternatively we tried random feature extension (Rahimi & Recht, 2008) for the position index $i$, and concatenated them to $\phi(x_i)$, however, the results were more or less the same.

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

# Appendix

We provide some formal derivations, experimental details and additional results that are skipped in the main paper.

## A  COMPLEXITY AND RUNNING TIME

In this section we analyze the complexity, especially the number of parameters and time complexity for a forward pass, of our DIEM. We also compare the wall clock running time for OTKE (Mialon et al., 2021) and our DIEM with different numbers of EM steps.

### A.1  COMPLEXITY

**Notation.** First, we recall the notation: The input set $S = \{x_1, \ldots, x_n\}$ contains $n$ $d$-dimensional feature vectors $x_i \in \mathbb{R}^d$. We consider the OTKE model with $p$ references $\{z_j\}_{j=1}^p$ (each $z_j \in \mathbb{R}^d$) as parameters. To be comparable, DIEM has $p$ Normal-Inverse-Wishart (NIW) parameters $\{\mu_j, \Sigma_j\}_{j=1}^p$ ($\mu_j \in \mathbb{R}^d$, $\Sigma_j$ is a $(d \times d)$ diagonal matrix), each of which defines a prior for each component $\mathcal{N}(z_j, V_j)$ in the Gaussian mixture. For simplicity, we assume single-head representations (i.e., $H = 1$) for both models, since multi-head ($H$ heads) representation increases all compute and memory linearly by $H$.

**Numbers of parameters.** The numbers of parameters are: $pd$ (OTKE) vs. $2pd$ (DIEM), Hence DIEM requires only twice as many parameters as OTKE.

**Time complexity.** First, one forward pass in OTKE consists of: [O1] computing the cost/kernel matrix $C$ which is $(n \times p)$ with $C_{ij} = -k(x_i, z_j)$, [O2] solving the OT problem (2) by the Sinkhorn-Knopp (SK) matrix scaling algorithm, and [O3] performing attention-weighted sums (1) to have the final embedding. The least amount of time for [O1] is $O(npd)$ to inspect each element of $x_i$ and $z_j$ once (e.g., linear kernels and squared-distance based kernels require $O(npd)$). Denoting the number of SK iterations by $M$, [O2] takes $O(npM)$, while [O3] requires $O(npd)$. Thus, the time complexity for one forward pass in OTKE is $O(np(M + d))$.

For DIEM with $k$ EM steps, [D1] each E-step (10) amounts to computing the Gaussian likelihoods, and additionally running the SK algorithm if the OT E-step is used. [D2] Each M-step (12) mainly consists of the weighted sums. [D1] takes $O(npd)$ for the regular E-step, and $O(np(M + d))$ for the OT E-step, while [D2] requires only $O(npd)$. Hence, the time complexity for one forward pass in our DIEM is $O(knp(M + d))$, only $k$ times as much as OTKE.

We summarize it in Table 6. The actual wall clock running times are compared in the next section.

### A.2  (FORWARD PASS) RUNNING TIME

To measure the forward pass time for OTKE and our DIEM, we consider first the large-scale SCOP 1.75 dataset: the input feature dimension $d = 512$ (from Gaussian RKHS) and the set cardinality $n = 1091$. To see the impact of the set cardinality, we inject random noise features to the sets to make twice and four times larger sets ($n = 2182$ and $n = 4364$). For fair comparison, both OTKE and DIEM use one head ($H = 1$) with the number of references (or mixture components) $p = 100$. The batch size is 128. We run all models on the same machine, Core i7 3.50GHz CPU and 128 GB RAM with a single GPU (RTX 2080 Ti).

Fig. 3 compares the average per-batch forward pass time for OTKE and DIEM with different EM steps ($k = 1$ to $4$). Although DIEM has an option to choose the E-step type (either regular E-step or OT E-step), we take all OT E-steps to have fair comparison with OTKE. (For the running time comparison between these two E-step types, please refer to Fig. 5.) As shown in Fig. 3, increasing the number of EM steps in our DIEM incurs only a small amount of computational overhead. The impact of the set cardinality $n$ on the running time exhibits similar trends for both OTKE and DIEM.

In Fig. 4, we compare running time of OTKE, SetTransformer (Lee et al., 2019a), and DIEM (with $k = 3$ EM steps). Here we use a minimal network structure for SetTransformer: one SAB block for the encoder, one PMA and one SAB block for the decoder. For larger sets ($\times 2$ and $\times 4$) we reduce

Table 6: Numbers of parameters and time complexity of OTKE (Mialon et al., 2021) and DIEM. We assume single-head models ($H = 1$). DIEM follows $k$ EM steps. Here, $n =$ input set cardinality, $d =$ input feature dimension, $p =$ number of references (OTKE) or mixture components (DIEM), and $M =$ number of SK iterations. Time complexity for one forward pass.

|  | OTKE | DIEM |
| --- | --- | --- |
| Number of parameters | $pd$ | $2pd$ |
| Time complexity | $O(np(M + d))$ | $O(knp(M + d))$ |

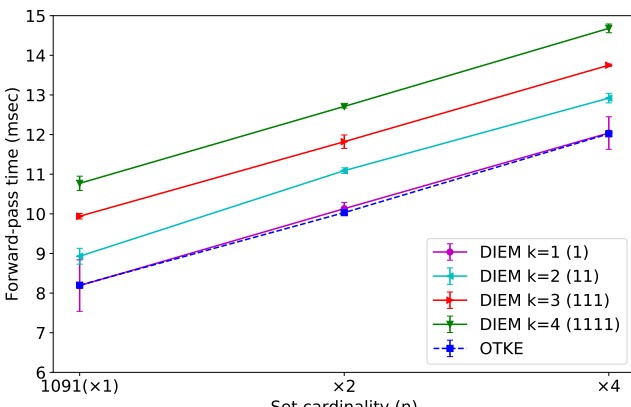

Figure 3: Average (per-batch) forward pass time (in milliseconds) for OTKE and DIEM with different EM steps ($k = 1$ to $4$) on the SCOP 1.75 dataset. In addition to the original set cardinality $n = 1091$ (denoted by $\times 1$ on the left corner), we increase $n$ by two ($\times 2$) and four times ($\times 4$). To be fair with OTKE, we use OT E-steps during the $k$ EM steps, denoted by sequences of 1's in the legend (e.g., $k = 4$ (1111) indicates that four OT E-steps are used during $k = 4$ EM steps).

the batch size for SetTransformer to fit into the GPU memory. We need not reduce the batch size for OTKE and DIEM as they fit into the GPU memory for all three set cardinalities.

The result shows that OTKE and DIEM are scalable to large sets whereas SetTransformer takes significantly longer time, which is well aligned with the fact that self-attention suffers from quadratic computational cost. In Fig. 5, we compare the running time between OT E-steps and regular E-steps in our DIEM. The maximum number of Sinkhorn-Knopp iterations for solving the OT problem is set to 10. While the regular E-step is consistently faster than the OT E-step, the increased overhead of the OT E-step seems to be minor compared to the overall running time.

For smaller datasets, we consider the OMNIGLOT counting (large set) and CIFAR100 clustering datasets. The running time of OTKE and DIEM on these two datasets is depicted in Fig. 6. For the OMNIGLOT large set, we set $p = 50, H = 2, d = 64$ for both models where $n \in \{11, \ldots, 30\}$ and batch size is 32 sets. For CIFAR-100 clustering, we test with $p = 4, H = 5, d = 512$ for both models where $n \in \{100, \ldots, 500\}$ and batch size is 10 sets. Our DIEM takes regular E steps. We vary the number of EM steps $k$ in DIEM from 1 to 5. The additional forward pass time incurred by extra EM steps in DIEM looks permissible, the overall running time of DIEM being constant factor comparable to that of OTKE.

## B  MATHEMATICAL DERIVATIONS

In Appendix B.1, we provide detailed derivations for MAP EM update equations (10-11) in the main paper. In Appendix B.2, we formally show that the inducing-point attention module of SetTransformer (Lee et al., 2019a) known as PMA, is actually a special case of OTKE (Mialon et al., 2021).

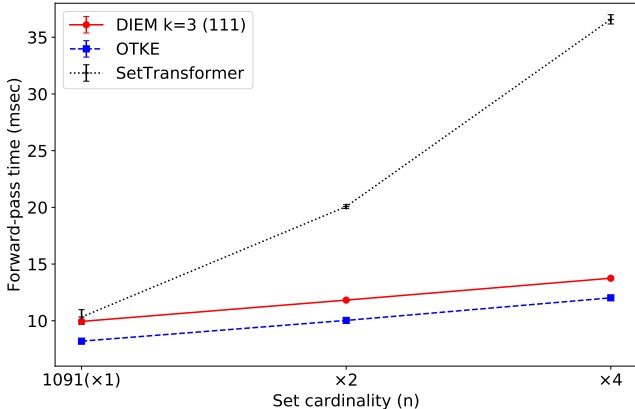

Figure 4: Running time comparison among OTKE, SetTransformer, and DIEM. Average (per-batch) forward pass time (in milliseconds) on the SCOP 1.75 dataset with three different set cardinalities (original, ×2, and ×4). DIEM takes three EM steps ($k = 3$ (111) indicates that three OT E-steps are used during $k = 3$ EM steps). SetTransformer uses a minimal network structure: one SAB block for the encoder, one PMA block and one SAB block for the decoder.

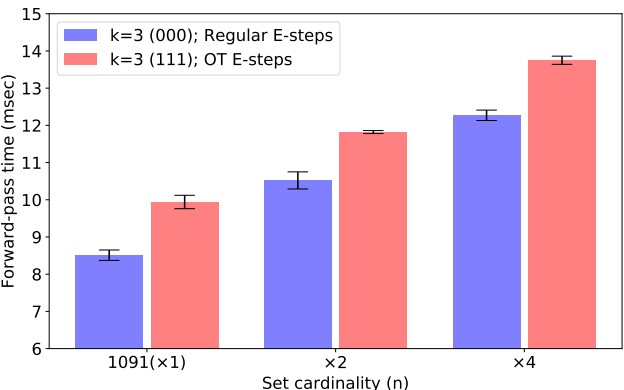

Figure 5: OT E-steps vs. regular E-steps in our DIEM. Average (per-batch) forward pass time on the SCOP 1.75 dataset with three different set cardinalities (original, ×2, and ×4). We compare DIEM with $k = 3$ (111) (all OT E-steps) with $k = 3$ (000) (all regular E-steps).

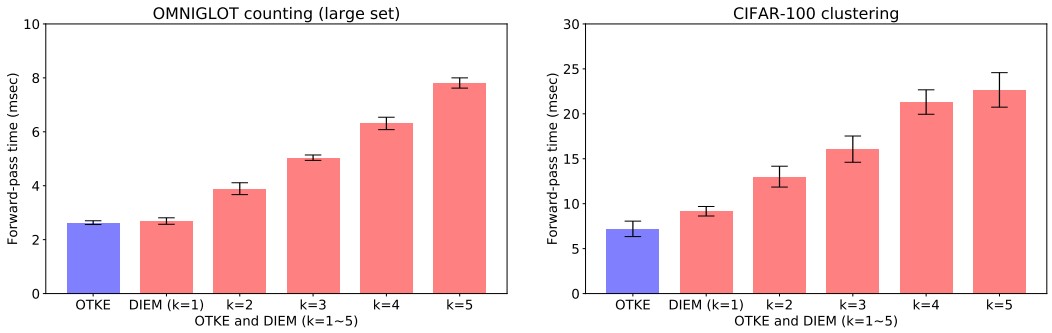

Figure 6: Running time comparison between OTKE and DIEM with $k = 1$ to $k = 5$ EM steps on (Left) OMNIGLOT counting (large set) and (Right) CIFAR-100 clustering.

## B.1 MAP EM Derivations

E-step is optimization of the lower bound in (9) with respect to $q(j|i)$ with $\theta$ fixed. Hence it is identical to the maximum likelihood E-step for the lower bound of (4). Now we derive the M-step, maximizing the lower bound in (9) with respect to $\theta$ with $q(j|i)$ fixed. That is,

$$\arg\max_\theta \ L(\theta) := \log p(\theta|\Lambda) + \sum_{ij} q(j|i) \log p(x_i, j). \tag{14}$$

The log-prior term of the Dir-NIW density can be expanded as ("$=_c$" refers to *equality up to constant*):

$$\log p(\theta|\Lambda) =_c -\frac{\nu+d+2}{2}\log|V_j| - \frac{\lambda}{2}(z_j-\mu_j)^\top V_j^{-1}(z_j-\mu_j) - \frac{1}{2}\mathrm{Tr}(\Sigma_j V_j^{-1})$$
$$+ (\eta-1)\sum_j \log \pi_j. \tag{15}$$

The weighted log-likelihood term in (14) can be written as:

$$\sum_{ij} q(j|i) \log p(x_i, j) =_c \sum_{ij} q(j|i)\left( \log \pi_j - \frac{1}{2}\log|V_j| - \frac{1}{2}(x_i-z_j)^\top V_j^{-1}(x_i-z_j) \right). \tag{16}$$

Combining (15) and (16), we take derivatives with respect to $\theta$ and solve for $\theta$ as follows:

$$\frac{\partial L}{\partial \pi_j} = \frac{\eta-1}{\pi_j} + \frac{\sum_i q(j|i)}{\pi_j} = \text{const} \implies \pi_j' = \frac{\sum_i q(j|i) + \eta - 1}{n + p \cdot (\eta-1)} \tag{17}$$

$$\frac{\partial L}{\partial z_j} = -\lambda V_j^{-1}(z_j-\mu_j) - V_j^{-1}\sum_i q(j|i)(z_j-x_i) = 0 \implies z_j' = \frac{\sum_i q(j|i)x_i + \mu_j\lambda}{\sum_i q(j|i) + \lambda} \tag{18}$$

$$\frac{\partial L}{\partial V_j^{-1}} = \frac{1}{2}\left( \Big(\nu+d+2+\sum_i q(j|i)\Big)V_j - \lambda(z_j-\mu_j)^2 - \Sigma_j - \sum_i q(j|i)(x_i-z_j)^2 \right) = 0$$

$$\implies V_j' = \frac{\sum_i q(j|i)(x_i-z_j)^2 + \Sigma_j + \lambda(z_j-\mu_j)^2}{\sum_i q(j|i) + \nu + d + 2} \tag{19}$$

$$= \frac{\sum_i q(j|i)(x_i^2 - (z_j')^2) + \Sigma_j + \lambda(\mu_j^2 - (z_j')^2)}{\sum_i q(j|i) + \nu + d + 2}, \tag{20}$$

where the equality between (19) and (20) can be easily verified from the optimal $z_j'$ in (18).

## B.2 PMA of SetTransformer is a Special Case of OTKE

The PMA (inducing-point attention) module of SetTransformer (Lee et al., 2019a) is an attention block that incorporates a fixed number of learnable parameters as queries, and plays an important role in SetTransformer to have a fixed size vector representation for a variable-length input set. Formally, PMA module has learnable $p$ vectors $Z = \{z_1, \ldots, z_p\}$, and for a given set $S = \{x_1, \ldots, x_n\}$, it performs the (dot-product) attention operation with each $z_j$ as a query and $S$ as keys and values. Hence each $z_j$ is transformed as follows:

$$z_j \rightarrow \sum_{i=1}^n \frac{\exp\big((W_q z_j)^\top(W x_i)/\sqrt{d}\big)}{\sum_{i=1}^n \exp\big((W_q z_j)^\top(W x_i)/\sqrt{d}\big)} W x_i, \tag{21}$$

where $W_q$ and $W$ are the weight matrices for queries and keys/values, respectively. Since $z_j$ itself is a learnable parameter vector, it is redundant to introduce $W_q$, and we can subsume it under $z_j$, which leads to

$$\textbf{PMA: } z_j \rightarrow \sum_{i=1}^n \frac{\exp\big(z_j^\top(W x_i)/\sqrt{d}\big)}{\sum_{i=1}^n \exp\big(z_j^\top(W x_i)/\sqrt{d}\big)} W x_i. \tag{22}$$

Next, we build a special OTKE module by having the RKHS feature mapping $\phi(x_i) = W x_i$, which corresponds to the linear kernel $k(x_i, z_j) = z_j^\top \phi(x_i) = z_j^\top W x_i$. Furthermore, we consider the OT

Table 7: (OMNIGLOT Small) Unique character counting on the OMNIGLOT small set. The averaged 0/1 accuracy results. In DIEM, $k = 2(00)$ (or $k = 1(0)$) means 2 (or 1) EM steps used both with regular E-steps.

| Method | Small set |
|---|---|
| DeepSet (Zaheer et al., 2017) | $0.4617 \pm 0.0076$ |
| Dot-prod Attn (Yang et al., 2018; Ilse et al., 2018) | $0.4471 \pm 0.0076$ |
| SetTransformer (SAB + PMA) (Lee et al., 2019a) | $0.6037 \pm 0.0075$ |
| OTKE ($p = 100, H = 1$) (Mialon et al., 2021) | $0.5754 \pm 0.0130$ |
| OTKE ($p = 50, H = 2$) (Mialon et al., 2021) | $0.5686 \pm 0.0187$ |
| OTKE ($p = 20, H = 3$) (Mialon et al., 2021) | $0.5216 \pm 0.0182$ |
| DIEM ($p = 20, H = 3, k = 1(0), \tau = 10^0$, SB) (Ours) | $0.6304 \pm 0.0056$ |
| DIEM ($p = 20, H = 3, k = 2(00), \tau = 10^{-3}$, SB) (Ours) | $0.6514 \pm 0.0074$ |
| DIEM ($p = 20, H = 3, k = 1(0), \tau = 10^0$, SB2) (Ours) | $0.6229 \pm 0.0114$ |
| DIEM ($p = 20, H = 3, k = 2(00), \tau = 10^{-3}$, SB2) (Ours) | $0.6665 \pm 0.0087$ |
| DIEM ($p = 20, H = 3, k = 1(0), \tau = 10^0$, PC) (Ours) | $0.6083 \pm 0.0194$ |
| DIEM ($p = 20, H = 3, k = 2(00), \tau = 10^{-3}$, PC) (Ours) | $\mathbf{0.7153 \pm 0.0067}$ |

problem between $Z$ and $S$ *without* the second constraints $\sum_{j=1}^{p} Q_{ij} = 1/n, \forall i$. That is,

$$\min_{Q} \sum_{ij} C_{ij} Q_{ij} - \epsilon H(Q) \text{ s.t. } \sum_{i=1}^{n} Q_{ij} = \frac{1}{p}, \ \forall j, \tag{23}$$

with $C_{ij} = -k(x_i, z_j) = -z_j^\top W x_i$. We can solve (23) using the Lagrange multiplier,

$$\mathcal{L} := \sum_{ij} C_{ij} Q_{ij} - \epsilon H(Q) - \sum_{j} \lambda_j \Big( \sum_{i} Q_{ij} - 1/p \Big). \tag{24}$$

We set the derivative to 0 and solve for $Q_{ij}$ as follows

$$\frac{\partial \mathcal{L}}{\partial Q_{ij}} = C_{ij} - \epsilon(\log Q_{ij} + 1) - \lambda_j = 0 \implies Q_{ij} = \exp\big( -C_{ij}/\epsilon + \alpha_j \big), \tag{25}$$

for some constant $\alpha_j$. Applying the constraints $\sum_{i=1}^{n} Q_{ij} = 1/p$ determines $\alpha_j$ and $Q_{ij}$ as:

$$e^{\alpha_j} = \frac{1}{p \sum_i \exp\big( -C_{ij}/\epsilon \big)}, \quad Q_{ij} = \frac{1}{p} \frac{\exp\big( z_j^\top (W x_i)/\epsilon \big)}{\sum_{i=1}^{n} \exp\big( z_j^\top (W x_i)/\epsilon \big)}. \tag{26}$$

By choosing the entropic regularizer trade-off $\epsilon = \sqrt{d}$ and applying the OTKE embedding formula $z_j \to \sqrt{p} \sum_i Q_{ij} \phi(x_i)$, we get

$$\textbf{OTKE:} \ z_j \to \frac{1}{\sqrt{p}} \sum_{i=1}^{n} \frac{\exp\big( z_j^\top (W x_i)/\sqrt{d} \big)}{\sum_{i=1}^{n} \exp\big( z_j^\top (W x_i)/\sqrt{d} \big)} W x_i, \tag{27}$$

which is equivalent to PMA (22) up to a constant factor. Hence PMA of SetTransformer can be viewed as an OTKE module without the equal element assignment constraints $\sum_{j=1}^{p} Q_{ij} = 1/n, \forall i$.

## C EXPERIMENTAL DETAILS AND ADDITIONAL RESULTS

### C.1 OMNIGLOT UNIQUE CHARACTER COUNTING

**(Original) Small set.** This experimental setup is the same as that of (Lee et al., 2019a). The sets of OMNIGLOT images are generated by: set cardinality $n \sim \{6, 7, 8, 9, 10\}$ and the number of unique characters $k \sim \{1, \ldots, n\}$. The batch is composed of 32 sets. The results are summarized in Table 7.

**Large set.** This large set experiment is built from the sets generated by: set cardinality $n \sim \{11, 12, \ldots, 30\}$ and the number of unique characters $k \sim \{5, \ldots, n\}$. The batch is composed of 32 sets. The results are shown in Table 8.

Table 8: (OMNIGLOT Large) Unique character counting on the OMNIGLOT large set. The averaged 0/1 accuracy results. In DIEM, $k = 2(00)$ means 2 EM steps used both with regular E-steps.

| Method | Large set |
|---|---|
| DeepSet (Zaheer et al., 2017) | $0.1927 \pm 0.0091$ |
| SetTransformer (ISAB32 + PMA) (Lee et al., 2019a) | $0.2456 \pm 0.0068$ |
| SetTransformer (ISAB64 + PMA) (Lee et al., 2019a) | $0.2836 \pm 0.0087$ |
| SetTransformer (SAB + PMA) (Lee et al., 2019a) | $0.3191 \pm 0.0050$ |
| OTKE ($p = 50, H = 2$) (Mialon et al., 2021) | $0.3352 \pm 0.0098$ |
| DIEM ($p = 50, H = 2, k = 2(00), \tau = 10^{-6}$, SB2) (Ours) | $0.3887 \pm 0.0071$ |
| DIEM ($p = 50, H = 2, k = 2(00), \tau = 10^{-6}$, PC) (Ours) | $0.4392 \pm 0.0056$ |
| DIEM ($p = 50, H = 2, k = 2(00), \tau = 10^{-3}$, PC) (Ours) | $\mathbf{0.4440 \pm 0.0069}$ |
| DIEM ($p = 50, H = 2, k = 2(00), \tau = 10^{0}$, PC) (Ours) | $0.3581 \pm 0.0121$ |

Table 9: Amortized clustering on synthetic data. The averaged test log-likelihood scores are shown. In DIEM, $k = 2(00)$ (or $k = 3(000)$) means 2 (or 3) EM steps used all with regular E-steps.

| Method | Synthetic |
|---|---|
| Oracle | -1.4726 |
| DeepSet (mean pooling) (Zaheer et al., 2017) | $-1.7606 \pm 0.0213$ |
| DeepSet (max pooling) (Zaheer et al., 2017) | $-1.7692 \pm 0.0130$ |
| Dot-prod Attn (Yang et al., 2018; Ilse et al., 2018) | $-1.8549 \pm 0.0128$ |
| SetTransformer (SAB + PMA) (Lee et al., 2019a) | $-1.5145 \pm 0.0046$ |
| SetTransformer (ISAB16 + PMA) (Lee et al., 2019a) | $-1.5009 \pm 0.0068$ |
| OTKE ($p = 4, H = 5$) (Mialon et al., 2021) | $-1.7803 \pm 0.0028$ |
| DIEM ($p = 4, H = 5, k = 3(000), \tau = 10^{-6}$, SB2) (Ours) | $-1.4883 \pm 0.0021$ |
| DIEM ($p = 4, H = 5, k = 3(000), \tau = 0.01$, SB2) (Ours) | $\mathbf{-1.4873 \pm 0.0018}$ |
| DIEM ($p = 4, H = 5, k = 3(000), \tau = 1$, SB2) (Ours) | $-1.4960 \pm 0.0039$ |
| DIEM ($p = 4, H = 5, k = 2(00), \tau = 0.01$, SB2) (Ours) | $-1.5082 \pm 0.0032$ |

**Network architectures and learning options.** The feature vectors $\phi(x)$ of images (set elements) are formed by applying four Conv(64,3,2,BN,ReLU) layers. For OTKE and our DIEM the output of the set embedding $emb(S)$ is fed into the fully-connected layers FC(64,ReLU) – FC(1,softplus) to return the (scalar) Poisson parameter. For SetTransformer $\phi(x)$ is fed into SAB(64,4) – SAB(64,4) – PMA(1,8,8) – FC(1,softplus) layers where SAB($d,h$) means the set attention block with $d/h$ units/-heads, and PMA($k,d,h$) is the Pooling Multihead Attention layer with $k$ vectors and $d/h$ units/heads. When the inducing-point attention is used, SAB(64,4) blocks are replaced by ISAB($m$,64,4) that has $m$ inducing points ($m = 32$ or $64$) with 64/4 units/heads. For DeepSet, $\phi(x)$ is fed into FC(64,ReLU) – FC(64,-) – Mean-Pool – FC(64,ReLU) – FC(1,softplus) layers. For all models, we use the Adam optimizer (Kingma & Ba, 2015) with learning rate $10^{-4}$ and batch size = 32 sets, until 200K iterations.

### C.2 Amortized Clustering

**Synthetic 2D data clustering.** The results are summarized in Table 9.

**Synthetic small set data clustering.** We test the models on sets with smaller cardinality: $N \sim \{50, \ldots, 100\}$ instead of the previous $N \sim \{100, \ldots, 500\}$. The results are shown in Table 10.

**CIFAR-100 clustering.** Results are shown in Table 11.

**Network architectures and learning options:** For OTKE and our DIEM, we first apply a linear layer FC($dh$,-) to the $din$-dimensional input vectors to have features $\phi(x)$, where $dh = 128$, $din = 2$ for the synthetic dataset and the small synthetic dataset, and $dh = 256$, $din = 512$ for the CIFAR-100. The output of the set embedding $emb(S)$ is then fed into FC($dh$,ReLU) – FC($dout$,-) where $dout = 4 + 4 \cdot din + 4$ for $\alpha$, $m$, and $C$, respectively, in the order-4 spherical-covariance Gaussian mixture parameter output $\Theta = \{\alpha, m, C\}$. We also apply softmax and softplus layers for $\alpha$ and $C$, respectively. For SetTransformer, the $din$-dimensional inputs are fed into SAB($dh$,4) – SAB($dh$,4) – PMA(4,$dh$,4) – SAB($dh$,4) – SAB($dh$,4) – FC($dout$,-) layers, whereas for the inducing-point

Table 10: Synthetic small set data clustering. The averaged test log-likelihood scores are shown. In DIEM, $k = 3(000)$ means 3 EM steps used all with regular E-steps.

| Method | Small Synthetic |
|---|---|
| DeepSet (Zaheer et al., 2017) | $-2.0955 \pm 0.0054$ |
| SetTransformer (ISAB16 + PMA) (Lee et al., 2019a) | $-1.5846 \pm 0.0077$ |
| SetTransformer (ISAB32 + PMA) (Lee et al., 2019a) | $-1.5277 \pm 0.0083$ |
| SetTransformer (SAB + PMA) (Lee et al., 2019a) | $-1.5099 \pm 0.0068$ |
| DIEM ($p = 4, H = 5, k = 3(000), \tau = 10^{-6}$, SB2) (Ours) | $-1.4297 \pm 0.0027$ |
| DIEM ($p = 4, H = 5, k = 3(000), \tau = 10^{-5}$, SB2) (Ours) | $-1.4308 \pm 0.0022$ |
| DIEM ($p = 4, H = 5, k = 3(000), \tau = 10^{-4}$, SB2) (Ours) | $-1.4274 \pm 0.0015$ |
| DIEM ($p = 4, H = 5, k = 3(000), \tau = 10^{-3}$, SB2) (Ours) | $-1.4301 \pm 0.0030$ |
| DIEM ($p = 4, H = 5, k = 3(000), \tau = 10^{-2}$, SB2) (Ours) | $\mathbf{-1.4253 \pm 0.0010}$ |
| DIEM ($p = 4, H = 5, k = 3(000), \tau = 10^{-1}$, SB2) (Ours) | $-1.4343 \pm 0.0015$ |

Table 11: Amortized clustering on CIFAR-100. The averaged test adjusted rand index (ARI) scores are shown (the higher the better). In DIEM, $k = 2(00)$ (or $k = 3(000)$) means 2 (or 3) EM steps used all with regular E-steps.

| Method | CIFAR-100 |
|---|---|
| Oracle | 0.9842 |
| DeepSet (mean pooling) (Zaheer et al., 2017) | $0.5736 \pm 0.0117$ |
| DeepSet (max pooling) (Zaheer et al., 2017) | $0.5463 \pm 0.0154$ |
| SetTransformer (SAB + PMA) (Lee et al., 2019a) | $0.9246 \pm 0.0113$ |
| SetTransformer (ISAB16 + PMA) (Lee et al., 2019a) | $0.9381 \pm 0.0122$ |
| OTKE ($p = 4, H = 5$) (Mialon et al., 2021) | $0.8207 \pm 0.0074$ |
| DIEM ($p = 4, H = 5, k = 2(00), \tau = 0.01$, SB2) (Ours) | $0.9709 \pm 0.0013$ |
| DIEM ($p = 4, H = 5, k = 3(000), \tau = 0.01$, SB2) (Ours) | $\mathbf{0.9770 \pm 0.0019}$ |
| DIEM ($p = 4, H = 5, k = 3(000), \tau = 0.01$, SB) (Ours) | $0.9688 \pm 0.0022$ |
| DIEM ($p = 4, H = 5, k = 3(000), \tau = 10^{-6}$, SB2) (Ours) | $0.9680 \pm 0.0011$ |
| DIEM ($p = 4, H = 5, k = 3(000), \tau = 1.0$, SB2) (Ours) | $0.9630 \pm 0.0024$ |

Table 12: SCOP 1.75 classification accuracies (top 1/5/10) for supervised learning. The hyperparameter $L$ in OTKE and DIEM indicates the number of Nyström anchor points for the approximation of Gaussian RKHS embedding. In DIEM, $k = 2(00)$ means 2 EM steps used both with regular E-steps. The hyperparameter $L$ is the number of anchor points used in the Nyström RKHS approximation of the feature $\phi(x)$ through the Gaussian kernel mapping on 10-mers (subsequences of length 10).

| Method | Supervised |
|---|---|
| DeepSF (Hou et al., 2019) | 73.0 / 90.3 / 94.5 |
| CKN (Chen et al., 2019a) | $84.1 \pm 0.1$ / $94.3 \pm 0.2$ / $96.4 \pm 0.1$ |
| RKN (Chen et al., 2019b) | $85.3 \pm 0.3$ / $95.0 \pm 0.2$ / $96.5 \pm 0.1$ |
| SetTransformer (Lee et al., 2019a) | $79.2 \pm 4.6$ / $91.5 \pm 1.4$ / $94.3 \pm 0.6$ |
| Rep-the-Set (Skianis et al., 2020) | $84.5 \pm 0.6$ / $94.0 \pm 0.4$ / $95.7 \pm 0.4$ |
| OTKE ($p = 50, H = 1, L = 128$) (Mialon et al., 2021) | $85.2 \pm 0.4$ / $94.5 \pm 0.2$ / $96.4 \pm 0.2$ |
| OTKE ($p = 10, H = 5, L = 512$) (Mialon et al., 2021) | $88.7 \pm 0.3$ / $95.9 \pm 0.2$ / $97.3 \pm 0.1$ |
| DIEM ($p = 50, H = 1, k = 2(00), \tau = 10^{-6}, L = 128$) (Ours) | $\mathbf{90.5 \pm 0.2}$ / $\mathbf{96.6 \pm 0.2}$ / $\mathbf{97.6 \pm 0.2}$ |
| DIEM ($p = 50, H = 1, k = 2(00), \tau = 10^{-6}, L = 512$) (Ours) | $89.6 \pm 0.3$ / $96.4 \pm 0.1$ / $97.6 \pm 0.1$ |

attention, the first two SAB($dh$,4) blocks are replaced by ISAB($m$,$dh$,4). For DeepSet, the layers are composed of $k_d$ FC($dh$,ReLU) layers, Mean-Pool, and $(k_d - 1)$ FC($dh$,ReLU), followed by FC($dout$,-), where $k_d = 4$ for synthetic and $k_d = 6$ for CIFAR-100. For all models, we use the Adam optimizer with learning rate $10^{-3}$ ($10^{-4}$ for CIFAR-100) and batch size = 10 sets, until 50K iterations.

## C.3 PROTEIN FOLD CLASSIFICATION TASK ON SCOP 1.75

**Unsupervised learning.** The best performing DIEM model is found with ($p = 100, H = 1, k = 1, \tau = 10^{-3}$) following the OT E-step. See also Fig. 12 for the results of different prior strength $\tau$ values.

**Supervised learning.** The results for several different hyperparameter choices are shown in Table 12.

**Network architectures and learning options.** We faithfully follow (Mialon et al., 2021) for the overall network architectures and learning options. For instance, the Adam optimizer is used with batch size 128 for 100 epochs, and the learning rate is initially 0.01 and halved if there is no decrease in validation loss for 5 consecutive epochs.

## C.4 EMPIRICAL STUDY ON HYPERPARAMETER IMPACTS

We empirically analyze the impact of various hyperparameters used in our DIEM model. They are summarized as follows.

**OMNIGLOT (small).** We first study the impact of prior strength $\tau$ and multi-head pooling strategy. While fixing the number of EM steps as $k = 2$, we vary $\tau \in \{10^{-6}, 10^{-3}, 10^0\}$ and the multi-head pooling strategy $\in \{PC, SB, SB2\}$. The results are shown in Fig. 7. We have the best accuracy when $\tau$ is the middle value $10^{-3}$ compared to the extreme ones ($10^{-6}$ lowest prior impact and $10^0$ highest impact). Regarding the multi-head pooling strategy, PC (parameter concatenation) performs the best for small $\tau$, while SB2 (select best with the selected mixture ID one hot encoding) is slightly better than PC for $\tau = 10^0$. Next we test the impact of the number of EM steps $k$ and the prior impact $\tau$. For this we fix the multi-head pooling strategy as SB2 or PC. As shown in Fig. 8, having multiple $k = 2$ EM steps consistently outperform a single EM step. Unlike $k = 2$, the single EM step models attained the best performance with the strongest prior impact $\tau = 10^0$.

**Synthetic clustering.** We also test the impact of multi-head pooling strategy and prior strength $\tau$ on this synthetic clustering dataset for our DIEM model ($p = 4, H = 5, k = 3$). As shown in Fig. 9, SB2 strategy outperforms the other two strategies for all $\tau$ values. For SB2, the impact of $\tau$ appears to be minor, while for the other two strategies, we have opposite behavior; larger $\tau$ performs better for PC, and smaller $\tau$ improves the performance for SB. Fig. 10 visualizes the impact of the number of EM steps, where the model is with ($p = 4, H = 5, \tau = 0.01$, SB2). It clearly shows that having multiple EM steps significantly improve the performance over the single EM step. There is only a little improvement from $k = 2$ to $k = 3$.

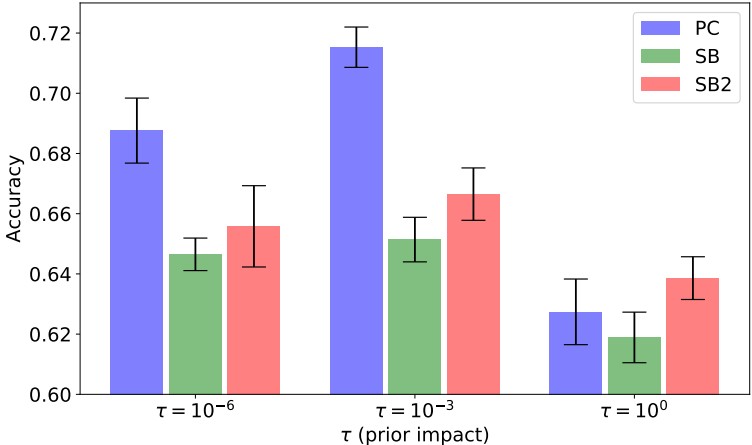

Figure 7: Impact of the prior strength hyperparameter $\tau$ and the multi-head pooling strategy for OMNIGLOT counting (small). The number of EM steps $k = 2$ fixed.

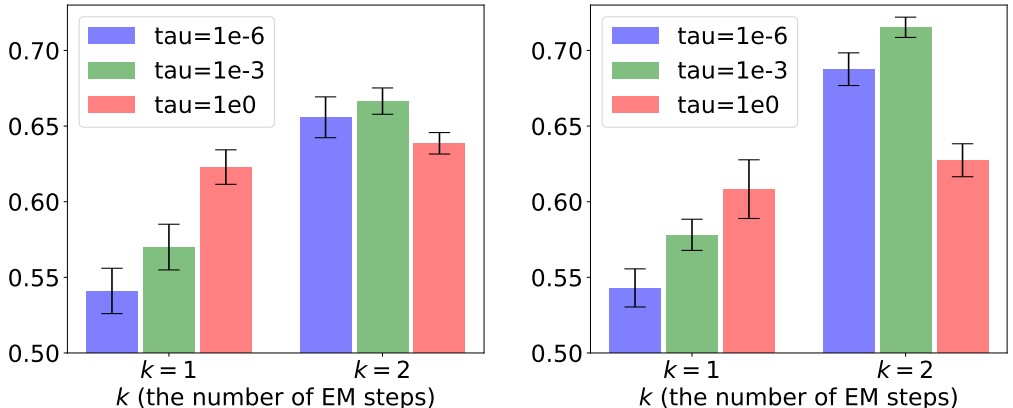

Figure 8: Impact of the number of EM steps $k$ and the prior impact $\tau$ for OMNIGLOT counting (small). The multi-head pooling strategy is fixed as SB2 (LEFT) or PC (RIGHT).

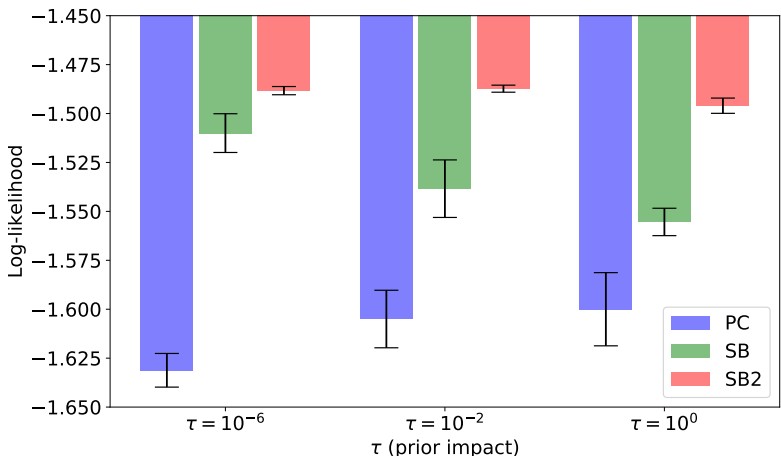

Figure 9: Impact of the prior impact hyperparameter $\tau$ and the multi-head pooling strategy for synthetic clustering.

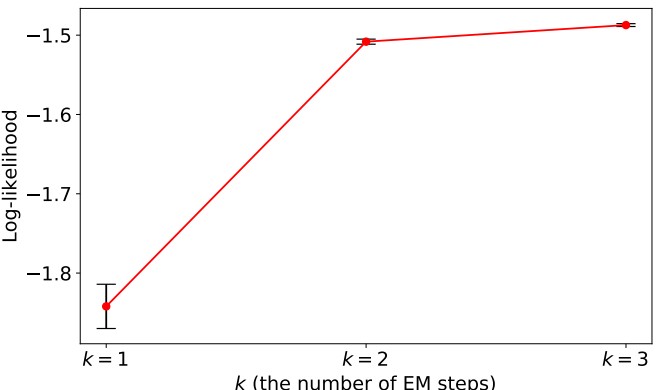

Figure 10: Impact of $k$ (the number of EM steps) for synthetic clustering. The model is with $(p = 4, H = 5, \tau = 0.01, \text{SB2})$.

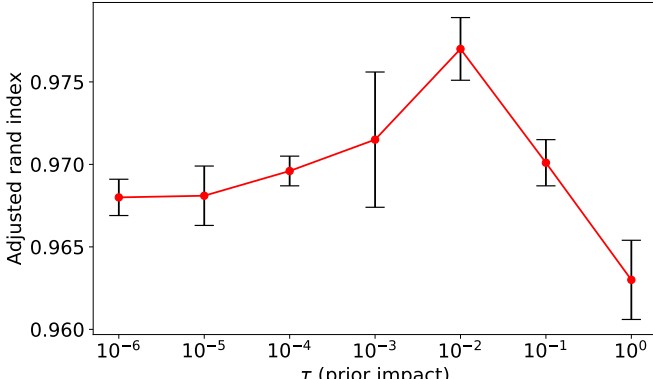

Figure 11: Impact of the prior strength hyperparameter $\tau$ on the CIFAR-100 clustering dataset with the DIEM model $(p = 4, H = 5, k = 3, \text{SB2})$.

**CIFAR-100 clustering.** We vary the prior impact hyperparameter $\tau$ from $10^{-6}$ to $10^{0}$, and the results are depicted in Fig. 11. It indicates that choosing a moderate prior trade-off $(10^{-2})$ yields the best accuracy.

**SCOP 1.75 protein fold classification.** We test on the protein fold classification dataset the impact of the prior strength $\tau$. For the unsupervised learning setting, we vary $\tau$ and the results are shown in Fig. 12. We have similar trend as CIFAR-100 clustering, where the moderate strength $\tau = 10^{-3}$ performs the best. Finally, we compare the types of E-steps performed for the supervised learning setup. For the DIEM model $(p = 50, H = 1, k = 2, \tau = 10^{-6}, L = 128)$, the $k = 2$ EM steps are run in two different ways: the regular E-steps and the OT E-steps (i.e., without and with imposing the balanced assignment constraints, respectively). The results are shown in Fig. 13, and the regular E-steps are more effective than OT E-steps for this particular dataset.

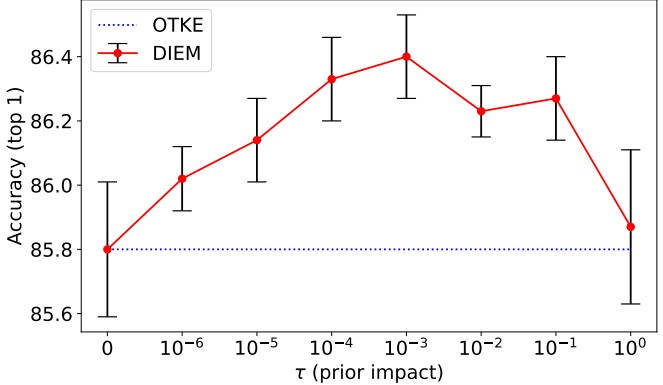

Figure 12: Impact of the prior strength hyperparameter $\tau$ on SCOP 1.75 unsupervised learning with the DIEM model ($p = 100, H = 1, k = 1(1)$).

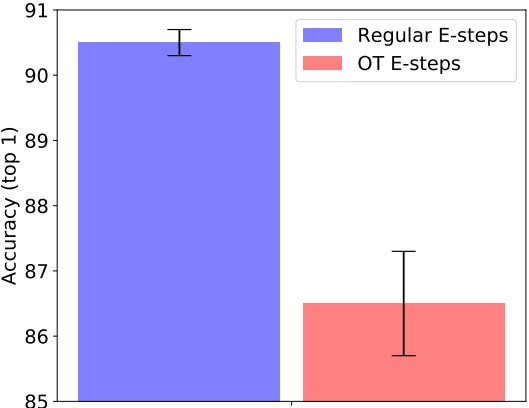

Figure 13: Comparison of regular E-steps and OT E-steps on SCOP 1.75 supervised learning.

