# OpenReview forum: "Differentiable Expectation-Maximization for Set Representation Learning"
_ICLR.cc/2022/Conference — ICLR 2022 Poster_

### Official Review · Reviewer_Y94U · 2021-10-28

**Correctness:** 3
**Technical Novelty And Significance:** 3
**Empirical Novelty And Significance:** 2
**Recommendation:** 8
**Confidence:** 3

**Details Of Ethics Concerns:**

None.

**Main Review:**

Pros:
- This work introduces a principled method for representing sets.
- The OTKE method is derived in a principled manner. An interesting consequence is that the choice of the number of reference can be made using the existing litterature of mixture fitting.
- Good experimental results on varied datasets (NLP, bioinformatics, vision, synthetic).
- Sensitivity studies for different hyperparameters.

Cons:
- The proposed method may somehow lack of novelty since the idea of using prototypes has been very studied recently.

Questions and remarks:
- What is the intuition of doing multiple EM steps in terms of embedding? Can this be related to the recent Perceiver [1] architecture? What is your view on this?
- Does DIEM learn the parameters of the prior distribution in the supervised setting? This could be more clear in the paper.
- The paper claims that the method has low computational complexity but it seems that this claim is not detailed in the paper (apart from remarks on the number of prototypes). Could you elaborate on the complexity of the EM steps?
- It could be great to provide more details on how to set the hyper-parameters for your method.
- Could you further discuss the impact of the prior depending on the task? Could we inject another prior/inductive bias here?
- Features given by protein language models such as ESM [2] can greatly improve results for SCOP 1.75. In fact, this may be the actual state-of-the-art for this dataset (see Table 5 in OTKE paper). Transfer learning is however orthogonal to the method proposed here but it is worth having this in mind.
- In the related work: "The limitation was found...": could you elaborate on this?

-----------------------

[1] Perceiver: General Perception with Iterative Attention (Andrew Jaegle and Felix Gimeno and Andrew Brock and Andrew Zisserman and Oriol Vinyals and Joao Carreira)

[2] Biological structure and function emerge from scaling unsupervised learning to 250 million protein sequences (Rives, Alexander and Meier, Joshua and Sercu, Tom and Goyal, Siddharth and Lin, Zeming and Guo, Demi and Ott, Myle and Zitnick, C. Lawrence and Ma, Jerry and Fergus, Rob)

**Summary Of The Paper:**

This work proposes a new embedding for sets of features, an important problem since many data modalities can be seen as such (images, sentences, etc.). More precisely, a set is represented by the output means of an EM algorithm for fitting the input set with a mixture of gaussians. The authors draw a new connection to an existing method for set embedding (OTKE). Moreover, their method achieves good experimental results.

**Summary Of The Review:**

The paper seems sound and provides new insights for set representation, with convincing experiments. I tend to recommend acceptance but it would be great if the authors could answer my questions.

---

> ### Author Response · Authors · 2021-11-19
> **Response to Reviewer Y94U**
>
> Thank you for your valuable feedback!
>
> **1. The idea of using prototypes has been very studied recently.**
>
> Although we use the prototypes as in OTKE and induced-set attention methods, the main contribution of our work is to discover the novel connection between prototype-based attention and mixture EM learning, and from this perspective, to generalize/extend existing approaches in reasonable/interesting ways.
>
>
> **2. Intuition of doing multiple EM steps in terms of embedding? Relation to the recent Perceiver [1] architecture?**
>
> Our intuition is that since OTKE is viewed as a Gaussian mixture learning with one-step EM, more EM iterations can have a better fit, hopefully better representing the set data. We appreciate the reviewer for referring to Perceiver [1]. Yes, applying multiple EM steps is architecturally similar to the Perceiver model that has multiple latent-attention layers (induced-set attention). We have added this reference in the revised paper (in Sec. 5).
>
> **3. Does DIEM learn the parameters of the prior distribution in the supervised setting? This could be more clear in the paper.**
>
> Yes. The output of the DIEM layer is fed into the subsequent layers including the final classification head, and the DIEM prior parameters (Lambda) can be updated in an end-to-end learning manner for supervised learning. We have made this statement more clear in the revised paper (the last paragraph of Sec. 4).
>
> **4. Computational complexity of the EM steps?**
>
> In the revised paper, the computational complexity of our DIEM (and OTKE) is analyzed in Appendix A.1 (summarized in Table 6). We also compare wall clock running times of DIEM, OTKE, and SetTransformers in Appendix A.2 (Fig. 3,4,5,6). Overall, DIEM is scalable to large sets, constant factor comparable to OTKE, and significantly faster than self-attention (SetTransformer).
>
> **5. More details on how to set the hyper-parameters for your method.**
>
> We think that the best hyperparameter choice strongly depends on the data characteristic. We can provide some general guidelines for selecting some hyperparameters in the following, but in practice we select them by cross validation as stated in the second paragraph of Sec. 6.
>
> About the pooling method (PC, SB, SB2) in multi-head representation: No single one is dominant for all datasets. A general guideline would be: 1) if the data exhibit multi-modal distributions and the types of distributions (e.g., numbers of modalities) are different across sets, then PC may be more adequate by having each head take charge of each type of distributions (e.g., OMNIGLOT counting); 2) if there is a single type of distributions across sets, then SB/SB2 is more effective since different heads aim to represent the same distribution (competing with each other), and the strategy of selecting the best-fit model would yield a better representation than enumerating all (e.g., CIFAR clustering).
>
> According to our results, it seems that the performance generally saturates with the number of EM steps ($k$), which also makes sense intuitively. However, the number of mixture components ($p$) needs to be carefully chosen (e.g., by validation) since it directly relates to overfitting/underfitting issues.
>
> **6. Could you further discuss the impact of the prior depending on the task? Could we inject another prior/inductive bias here?**
>
> Generally we can say that for datasets with large set cardinalities, the small prior impact leads to better performance (e.g., synthetic clustering, SB in Fig. 9 of the revised paper), while for datasets with small set cardinalities (e.g., OMNIGLOT counting), larger prior impact improves the performance over smaller one (Fig. 7 in the revised paper, up to $\tau=10^{-3}$). This empirical result is also well aligned with the intuition. One could in principle introduce a hierarchical prior or inductive bias as the reviewer suggested. In practice we select hyperparameters by validation.
>
> **7. Features given by protein language models such as ESM [2] can greatly improve results for SCOP 1.75. In fact, this may be the actual state-of-the-art for this dataset (see Table 5 in OTKE paper). Transfer learning is however orthogonal to the method proposed here but it is worth having this in mind.**
>
> Thank you for sharing the insight. We will keep this in mind.
>
> **8. In the related work: "The limitation was found...": could you elaborate on this?**
>
> The following excerpt from (Wagstaff et al.) states the limitation of DeepSets.
> *(DeepSets have a functional form of summation of latent features.) In particular, it has been conjectured (in DeepSets) that the dimension of the latent space may remain fixed as the cardinality of the sets under consideration increases. However, ... the analysis leading to this conjecture requires mappings which are highly discontinuous and ... this is only of limited practical use ...*

---

> > ### Comment · Reviewer_Y94U · 2021-11-22
> > **Thank you!**
> >
> > Thank you for your clarifications. In addition to clear motivation and good experimental results, and although a slight lack of novelty, I think this work provides nice insights on OTKE and Perceiver. Hence, I will increase my score.

---

### Official Review · Reviewer_xBWk · 2021-11-02

**Correctness:** 4
**Technical Novelty And Significance:** 3
**Empirical Novelty And Significance:** 3
**Recommendation:** 6
**Confidence:** 2

**Main Review:**

Strengths:
1. The connections between OTKE and EM is insightful in set representation learning, and differentiable EM is well motivated.
2. Experimental results are impressive and support the claims made in this paper well.

Weakness:
1. Time complexity or empirical wall-clock time is needed to give a thorough analysis of differentiable EM. It will be helpful to present the time complexity (or empirical wall-clock time) of differentiable EM, since it takes several EM steps and costs more time compared to OTKE.


**Summary Of The Paper:**

This paper discusses that optimal transport kernel embedding (OTKE) can be regarded as a single expectation-maximization (EM) step towards the maximum likelihood estimate of Gaussian mixture models under mild conditions. Motivated by the finding, this paper proposes differentiable EM, which can be regarded as a generalized version of OTKE with prior and several EM steps. Experiments on OMNIGLOT unique character counting, amortized clustering in CIFAR-100, protein fold classification on SCOP 1.75, sentiment classification on SST-2 and chromatin profile detection on deepsea demonstrate the effectiveness of differentiable EM on set representation learning.

**Summary Of The Review:**

This paper presents a novel idea about set representation learning. Experiments cover multiple tasks and support the claims well. Though more analysis on time complexity is needed, I think this paper is above the acceptance threshold.

---

> ### Author Response · Authors · 2021-11-19
> **Response to Reviewer xBWk**
>
> Thank you for your valuable feedback!
>
> **1. Time complexity or empirical wall-clock time is needed to give a thorough analysis of differentiable EM.**
>
> In the revised paper, the computational complexity of our DIEM (and OTKE) is analyzed in Appendix A.1 (summarized in Table 6). We also compare wall clock running times of DIEM, OTKE, and SetTransformers in Appendix A.2 (Fig. 3,4,5,6). Overall, DIEM is scalable to large sets, constant factor comparable to OTKE, and significantly faster than self-attention (SetTransformer).

---

### Official Review · Reviewer_emDK · 2021-11-02

**Correctness:** 3
**Technical Novelty And Significance:** 3
**Empirical Novelty And Significance:** 3
**Recommendation:** 6
**Confidence:** 4

**Main Review:**

Overall, I like the paper; it is well written, and the interpretation of the set-embedding procedure as an EM iteration indeed makes sense. It is also good to see the authors derive a novel algorithm from their re-interpretation. The experiments are diverse and thorough, and as far as I can see, they seem to be reproducible with all the details provided in the appendix.

I think the paper can be enhanced with some further clarification.

1) In my opinion, it is quite important to compare the number of parameters when comparing different set embedding methods; for instance, in (Lee et al., 2019), they set the number of parameters for DeepSets and Set transformers roughly the same. How many parameters were used for the proposed method? I hope to see the parameter counts at least in the appendix. It would also be helpful to compare the wall-clock time for the forward passes; especially, for the proposed method, it is worth checking the inference time w.r.t. the number of EM iterations $k$.

2) There are quite a few hyperparameters or options for the proposed model; the number of mixture components $p$, number of EM iterations $k$, prior hyperparameter $\tau$, and the way of pooling (PC, SB, or SB2). Judging from the appendix, the performance of the proposed approach is quite sensitive to the choice of these hyperparameters. I'm also quite confused with three options for the pooling; is there any guide for which one to choose? Was any of those three pooling methods dominant in general? It is quite hard to directly compare the effect of individual choices of the hyperparameters because the results so far is not controlled experiments for the hyperparameters. Does the performance generally saturate with the number of mixture components $p$ or the number of EM steps $k$?

3) Have you considered using generative models other than a mixture of Gaussians? I guess the primary reason for the choice is its conjugacy, but probably we can think of other conjugate pairs for the mixture components.

4) Collapsing the hyperparameters $\tau = \eta-1 = \lambda = 1 = \nu + d + 2$ is weird; for instance, $\nu + d + 2$ cannot be equal to one. Can you elaborate on this?

5) How important is the step to initialize the parameters as the mode of the posteriors? What happens with the randomly initialized parameters or learning them as well with gradient descent? For instance, if the mixture components are not conjugate so the MAP parameters are not easily estimated, then we may consider different options.



**Summary Of The Paper:**

This paper proposes a novel set embedding method inspired by the EM algorithm. Treating each element in a set as i.i.d. samples from a mixture of Gaussians, the procedure of computing pairwise similarities between the elements and prefixed set of reference vectors corresponds to the computation of responsibilities in E-step for the mixture of Gaussians, and the embedding step using the similarities corresponds to the parameter update in M-step. The previous approaches such as OTKE can directly be interpreted with this EM view (plus balanced assignment constraint). Based on this reinterpretation, the paper proposes a novel set-embedding method extending previous methods in various ways; 1) use multiple steps of EM updates, 2) learn parameters other than reference vectors (covariances and mixing proportions), 3) learn the initial value of the parameters by placing prior distributions on them. The resulting algorithm entitled DIfferentiable EM (DIEM) is demonstrated to excel in various set-to-vec tasks.

**Summary Of The Review:**

The paper proposes an interesting idea, and the experimental results are promising. There are some minor concerns to be clarified.

---

> ### Author Response · Authors · 2021-11-19
> **Response to Reviewer emDK**
>
> Thank you for your valuable feedback!
>
> **1. Number of parameters and wall-clock time (with respect to the number of EM steps).**
>
> In the revised paper, the computational complexity of our DIEM (and OTKE) is analyzed in Appendix A.1 (summarized in Table 6). We also compare wall clock running times of DIEM, OTKE, and SetTransformers in Appendix A.2 (Fig. 3,4,5,6). Overall, DIEM is scalable to large sets, constant factor comparable to OTKE, and significantly faster than self-attention (SetTransformer).
>
> **2. Hyperparameters (e.g., number of mixture components, number of EM iterations, pooling method)**
>
> About the pooling method (PC, SB, SB2) in multi-head representation, we think that the best choice strongly depends on the data characteristic. No single one is dominant for all datasets. A general guideline would be: 1) if the data exhibit multi-modal distributions and the types of distributions (e.g., numbers of modalities) are different across sets, then PC may be more adequate by having each head take charge of each type of distributions (e.g., OMNIGLOT counting); 2) if there is a single type of distributions across sets, then SB/SB2 is more effective since different heads aim to represent the same distribution (competing with each other), and the strategy of selecting the best-fit model would yield a better representation than enumerating all (e.g., CIFAR clustering).
>
> According to our results, it seems that the performance generally saturates with the number of EM steps ($k$), which also makes sense intuitively. However, the number of mixture components ($p$) needs to be carefully chosen (e.g., by validation) since it directly relates to overfitting/underfitting issues.
>
>
> **3. Have you considered using generative models other than a mixture of Gaussians? I guess the primary reason for the choice is its conjugacy, but probably we can think of other conjugate pairs for the mixture components.**
>
> Yes. In principle, any generative model can be used: $z$ as parameters (weights) of the generative model that aims to represent data (set). With the choice of Gaussian mixture and EM, we have a close-form update equation. Of course, distributions other than Gaussian can be used if conjugacy is preserved.
>
> **4. Collapsing the hyperparameters $\tau = \eta-1 = \lambda = \nu+d+2$ is weird; for instance, $\nu+d+2$ cannot be equal to one. Can you elaborate on this?**
>
> There is no exact equivalence/correspondence between the MAP solution (11) and the simplified one (12) as we collapse the hyperparameters into one. We aim to reduce the number of hyperparametrs, at the expense of having update equations that slightly stray off from the original ones.
>
>
> **5. How important is the step to initialize the parameters as the mode of the posteriors? What happens with the randomly initialized parameters or learning them as well with gradient descent?**
>
> It is crucial to initialize the Gaussian mixture parameters in EM by the mode of the prior. We tried with random initials, and it didn't work well. There are several explanations for this: 1) As the mixture likelihood maximization problem is non-convex, it is sensitive to where we start EM. The mode of the prior is perhaps the most reasonable choice (when EM step $k=0$, the mode of the prior is the optimal solution). 2) By initializing with the mode of the prior, we can also establish strong dependency between the final EM solution output $\theta^k$ and the model parameters (prior's $\Lambda$), which also helps strengthening the learning signals for the $\Lambda$ update in backpropagation. Otherwise, $\Lambda$ only affects $\theta^k$ through $\tau$-scaled terms in the numerator in Eq(12). 3) In the extreme case where $\tau$ vanishes (prior impact is zero, i.e., ML-EM), the dependency of the output $\theta^k$ on $\Lambda$ is solely established by the mode-based initialization, and random initialization would make the two independent, thus $\Lambda$ would not be updated at all.

---

### Official Review · Reviewer_7HuD · 2021-11-05

**Correctness:** 3
**Technical Novelty And Significance:** 3
**Empirical Novelty And Significance:** 3
**Recommendation:** 6
**Confidence:** 2

**Main Review:**

The paper is well written and easy to understand. However, I do have some comments:
1) It is obvious to see that DIEM achieves better results than OTKE baseline in terms of offline evaluation metrics, such as accuracy, log-likelihood score. And as the author mentioned, the improvements come from the multiple steps EM algorithms. If this is the case, has the runtime been increased? In addition, the author also mentioned that OTKE-type methods would reduce the computational cost compared with attention (Set)Transformer. Based on two arguments, the running time probably should be compared between different baselines.

2)  DIEM doesn't have better results than OTKE on the largest DeepSEA dataset, which would influence the practical performance of DIEM on the large-scale NLP/Bioinformatics tasks.



**Summary Of The Paper:**

The paper is well written. In this paper, the author proposed an EM-based algorithm, DIEM, for set representation learning. The author first provides the equivalence between the OTKE representation learning algorithm and a single-step EM algorithm with extra balanced assignment constraints on the E-step. Then DIEM is developed and consistently outperforms/competes with OTKE algorithms in different empirical studies with the assistance of multiple EM steps and extra regularization. And DIEM is applicable both for supervised and unsupervised settings.

**Summary Of The Review:**

Please refer above.

---

> ### Author Response · Authors · 2021-11-19
> **Response to Reviewer 7HuD**
>
> Thank you for your valuable feedback!
>
> **1. Running time comparison**
>
> In the revised paper, the computational complexity of our DIEM (and OTKE) is analyzed in Appendix A.1 (summarized in Table 6). We also compare wall clock running times of DIEM, OTKE, and SetTransformers in Appendix A.2 (Fig. 3,4,5,6). Overall, DIEM is scalable to large sets, constant factor comparable to OTKE, and significantly faster than self-attention (SetTransformer).
>
>
> **2. Practical performance of DIEM on the large-scale NLP/Bioinformatics tasks.**
>
> For the SST-2 and DeepSEA tasks, it is true that there is less significant impact of the higher-level attention (OTKE/DIEM) layers, whereas the features at low-level layers seem to be more crucial (e.g., the 1D conv layers in DeepSEA). However, on the other large-scale task, SCOP 1.75, the improvement of DIEM over OTKE via multi-step EM updates does look significant.

---

### Author Response · Authors · 2021-11-19
**Revised manuscript with additional experiments and clarifications**

We thank all reviewers for their insightful and constructive comments/questions. Below we summarize the revisions made to the manuscript reflecting reviewers' comments. Our responses to reviewers' individual comments/questions are placed after each review.

**1. Computational complexity, number of parameters, and wall-clock running times**

In the revised paper, the computational complexity and the numbers of parameters of our DIEM and OTKE are analyzed in Appendix A.1 (summarized in Table 6). We also compare wall clock running times of DIEM, OTKE, and SetTransformers in Appendix A.2 (Fig. 3,4,5,6). Overall, DIEM is scalable to large sets, constant factor comparable to OTKE, and significantly faster than self-attention (SetTransformer).

**2. Clarification and citation**

We have revised the last paragraph of Sec. 4 to clarify the use of DIEM in supervised and unsupervised settings. A new citation for the Perceiver model (Jaegle et al.) has been added to Sec. 5 (Related Work),
whose multiple latent-attention layers are architecturally similar to applying multiple EM steps in our DIEM. We thank Reviewer Y94U for pointing them out.

---

### Decision · Program_Chairs · 2022-01-20

**Decision:**

Accept (Poster)

**Comment:**

This work proposes a new embedding for sets of features. A set is represented by the output means of an EM algorithm for fitting the input set with a mixture of Gaussians. The authors draw a new connection to an existing method for set embedding (OTKE). Moreover, their method achieves good experimental results.

There is general consensus among the reviewers that the paper is sound, well-written and provides new insights for set representation, with convincing experiments.

The authors have answered to most comments raised by the reviewers and have revised the paper accordingly.

I recommend acceptance as a poster.